# USP12 downregulation orchestrates a protumourigenic microenvironment and enhances lung tumour resistance to PD-1 blockade

Zhaojuan Yang [1,8], Guiqin Xu[1,8], Boshi Wang[1,8], Yun Liu[1], Li Zhang[1], Tiantian Jing[1], Ming Tang[1], Xiaoli Xu[2], Kun Jiao[2], Lvzhu Xiang[1], Yujie Fu[3], Daoqiang Tang[4], Xiaoren Zhang[5], Weilin Jin [6], Guanglei Zhuang [1,7], Xiaojing Zhao [3✉] & Yongzhong Liu [1✉]

Oncogenic activation of KRAS and its surrogates is essential for tumour cell proliferation and survival, as well as for the development of protumourigenic microenvironments. Here, we show that the deubiquitinase USP12 is commonly downregulated in the $Kras^{G12D}$-driven mouse lung tumour and human non-small cell lung cancer owing to the activation of AKT-mTOR signalling. Downregulation of USP12 promotes lung tumour growth and fosters an immunosuppressive microenvironment with increased macrophage recruitment, hypervascularization, and reduced T cell activation. Mechanistically, USP12 downregulation creates a tumour-promoting secretome resulting from insufficient PPM1B deubiquitination that causes NF-κB hyperactivation in tumour cells. Furthermore, USP12 inhibition desensitizes mouse lung tumour cells to anti-PD-1 immunotherapy. Thus, our findings propose a critical component downstream of the oncogenic signalling pathways in the modulation of tumour-immune cell interactions and tumour response to immune checkpoint blockade therapy.

[1] State Key Laboratory of Oncogenes and Related Genes, Shanghai Cancer Institute, Renji Hospital, Shanghai Jiao Tong University School of Medicine, Shanghai, China. [2] Shanghai Jiao Tong University School of Biomedical Engineering, Shanghai, China. [3] Department of Thoracic Surgery, Ren Ji Hospital, School of Medicine, Shanghai Jiao Tong University, Shanghai, China. [4] Department of Pathology, Ren Ji Hospital, School of Medicine, Shanghai Jiao Tong University, Shanghai, China. [5] Affiliated Cancer Hospital and Institute, Guangzhou Medical University, Guangzhou, China. [6] Institute of Cancer Neuroscience, Medical Frontier Innovation Research Center, The First Hospital of Lanzhou University, The First Clinical Medical College of Lanzhou University, Lanzhou, China. [7] Shanghai Key Laboratory of Gynecologic Oncology, Ren Ji Hospital, Shanghai Jiao Tong University School of Medicine, Shanghai, China. [8] These authors contributed equally: Zhaojuan Yang, Guiqin Xu, Boshi Wang. ✉email: drzhaoxiaojing@aliyun.com; liuyzg@shsci.org

Tumours are complex tissues consisting of multiple types of cells. The extensive interactions among different cellular populations in local tumours determine tumour behaviour and malignant progression. Aberrations in certain signalling pathways within tumour cells not only drive tumour cell proliferation and survival but also dictate the development of pro-tumourigenic microenvironments[1,2]. Oncogenic activation of KRAS in tumour cells induces chemokine expression to facilitate cell proliferation[3], tumour vascularization, and myeloid cell infiltration[4,5]. Similarly, the loss of PTEN leads to reduced numbers and impaired function of tumour-infiltrating T cells by increasing the expression of immunosuppressive cytokines[6]. Myc signalling controls IL-23 and CCL9 expression to inhibit the recruitment of T, B, and NK cells but increases macrophage infiltration and angiogenesis in a $Kras^{G12D}$-driven lung adenoma mouse model[7]. The oncogenic events also increase PD-L1 expression and confer immunoresistance to tumour cells[8–10]. Therefore, cancer cell-intrinsic oncogenic events function as critical determinants for the development of the tumour microenvironment (TME).

Lung cancer is the most common cause of mortality worldwide[11,12]. The oncogenic mutation of tumour-inducing genes, such as KRAS and EGFR or their surrogates, is prevalent in human lung cancers. In lung adenocarcinoma (LUAD), KRAS and EGFR mutations are found in approximately 32% and 27% of patients, respectively, whereas mutations in PTEN and PI3CA are present in 15–16% of lung squamous cell carcinomas (LUSCs)[11]. Defining the convergent molecules underlying the oncogenic events-mediated regulation of TME may help establish new therapeutic interventions for lung cancer.

Deubiquitinases (DUBs) are key regulators of ubiquitin-mediated signalling pathways. The dysregulation of certain DUBs, due to either genetic mutation or differential abundance, plays an important role in tumour development and progression[13–15]. Consequently, DUBs are emerging as attractive pharmacological targets for the development of novel anticancer drugs[14]. USP12, a member of the ubiquitin-specific protease (USP) family, has been reported to act as a cysteine protease to deconjugate ubiquitin from several substrates, including PHLPP1[16], histones H2A/H2B[17], NOTCH[18], AR[19], and MDM2[20]. USP12 also non-catalytically induces cell autophagy in Huntington's disease[21]. However, the clinical relevance of USP12 in human tumour development and progression, as well as its potential impacts on tumour immunotherapy, remain elusive.

Here, we show that USP12 expression is downregulated in $Kras^{G12D}$-driven mouse lung tumour and human non-small cell lung cancer (NSCLC). Downregulation of USP12 contributes to the development of the protumourigenic microenvironment, which is mainly characterized by increased macrophage recruitment, augmented angiogenesis, and decreased T cell activation, through regulating protumourigenic chemokine expression. Furthermore, USP12 inhibition substantially induces mouse tumour resistance to PD-1 blockade. Thus, our results illustrate that USP12, as a convergent regulator downstream of tumour-driven events in NSCLC, plays a role in reprogramming the TME and influences the tumour response to immune checkpoint blockade.

## Results

**USP12 expression is downregulated in NSCLC.** We initially set out to examine the alterations in the transcriptional profiles of DUBs in response to KRAS activation. Mouse embryonic fibroblasts (MEFs) with or without the Loxp-Stop-Loxp-$Kras^{G12D}$ allele were immortalized with the SV40 large T antigen (SV40-LT) and subsequently modified by Cre recombinase-induced

activation of oncogenic $Kras^{G12D}$. We compared the transcriptional profiles of DUBs between tumourigenic $Kras^{G12D}$ MEFs ($Kras^{G12D}$;SV40-LT MEFs) and non-tumourigenic $Kras^{wt}$ MEFs ($Kras^{wt}$;SV40-LT MEFs) using RNA sequencing (RNA-seq). This initial screen identified 6 DUB genes, Usp25, Usp12, Usp13, Cyld, Usp53, and Yod1, with dramatically decreased expression in $Kras^{G12D}$; SV40-LT MEFs (Supplementary Fig. 1a). We further analysed the expression patterns of these genes in human NSCLC using the TCGA-LUAD and GSE31210 databases and found that only USP12 was preferentially downregulated in both NSCLC tumours and tumourigenic $Kras^{G12D}$;SV40-LT MEFs (Fig. 1a and Supplementary Fig. 1b). Thus, these independent and unbiased analyses pointed to the potential role of USP12 in regulating the development of NSCLC. As expected, markedly lower protein expression of USP12 was observed in $Kras^{G12D}$;SV40-LT MEFs than in control cells (Fig. 1b). In the mouse $Kras^{G12D}$-driven lung tumour model, we further confirmed USP12 downregulation at both the mRNA and protein levels in tumour specimens compared with normal adjacent tissues (Fig. 1c, d). To test whether USP12 downregulation is strictly associated with KRAS mutation in NSCLC, we analysed USP12 transcript levels in human LUAD tumours carrying different types of oncogenic mutations (TCGA-LUAD database). We found that decreased expression of USP12 was present in tumours with different driver gene mutations (Fig. 1e), suggesting that the downregulation of USP12 is not a phenomenon specifically linked to the KRAS mutant. In addition to human LUADs, decreased expression of USP12 was found in LUSC tumours (Supplementary Fig. 1c). The downregulated USP12 transcription in tumours was also demonstrated in other NSCLC databases and broadly present in the tumours of the patients at different stages (Fig. 1f and Supplementary Fig. 1d). Accordingly, the overwhelming majority of tumour tissues from NSCLC patients (17/18 cases) exhibited lower protein levels of USP12 than the corresponding non-tumour tissues (Fig. 1g). In addition, USP12 expression was negatively associated with poor prognosis in NSCLC patients (Fig. 1h). Overall, these results reveal a significant prevalence of USP12 downregulation in human NSCLC and suggest that USP12 downregulation may represent a convergent response to the signals driven by oncogenic mutations or their surrogates in tumour cells.

**USP12 expression is inhibited by AKT-mTOR signalling.** Next, we sought to investigate which signalling pathway was mainly responsible for USP12 downregulation in tumour cells by utilizing the inhibitors targeting the main pathways downstream of the oncogenic mutations or constitutively activated in NSCLC. Treatment with inhibitors targeting MEK (U0126), ERK (ERK inhibitor), JNK (JNK inhibitor II), JAK-STAT3 (WP1066), c-Myc (c-Myc inhibitor), p38 MAPK (SB203580), and TGF-β-SMAD (SB431542 and SIS3) failed to produce a substantial change in USP12 expression (Supplementary Fig. 1e); however, blocking AKT-mTOR signalling using the selective AKT inhibitor API-2 or the mTOR inhibitor rapamycin significantly increased USP12 expression (Fig. 1i, j). Consistently, ectopic expression of the myristoylated form of AKT1 (myr-AKT1) significantly repressed USP12 expression in both human and mouse tumour cells (Fig. 1k); knockdown of endogenous AKT1 or AKT2, the two major isoforms of AKT, induced higher levels of USP12 than did the control treatment (Fig. 1l). In accordance, both API-2 and Rapamycin significantly increased the luciferase activities driven by the USP12 promoter (Supplementary Fig. 1f, g), underscoring a transcriptional regulation taking place in the control of USP12 expression by AKT-mTOR signalling. Deletion analysis of USP12 promoter activity showed that the region spanning −2928 to −1337 bases relative to the transcriptional start site was

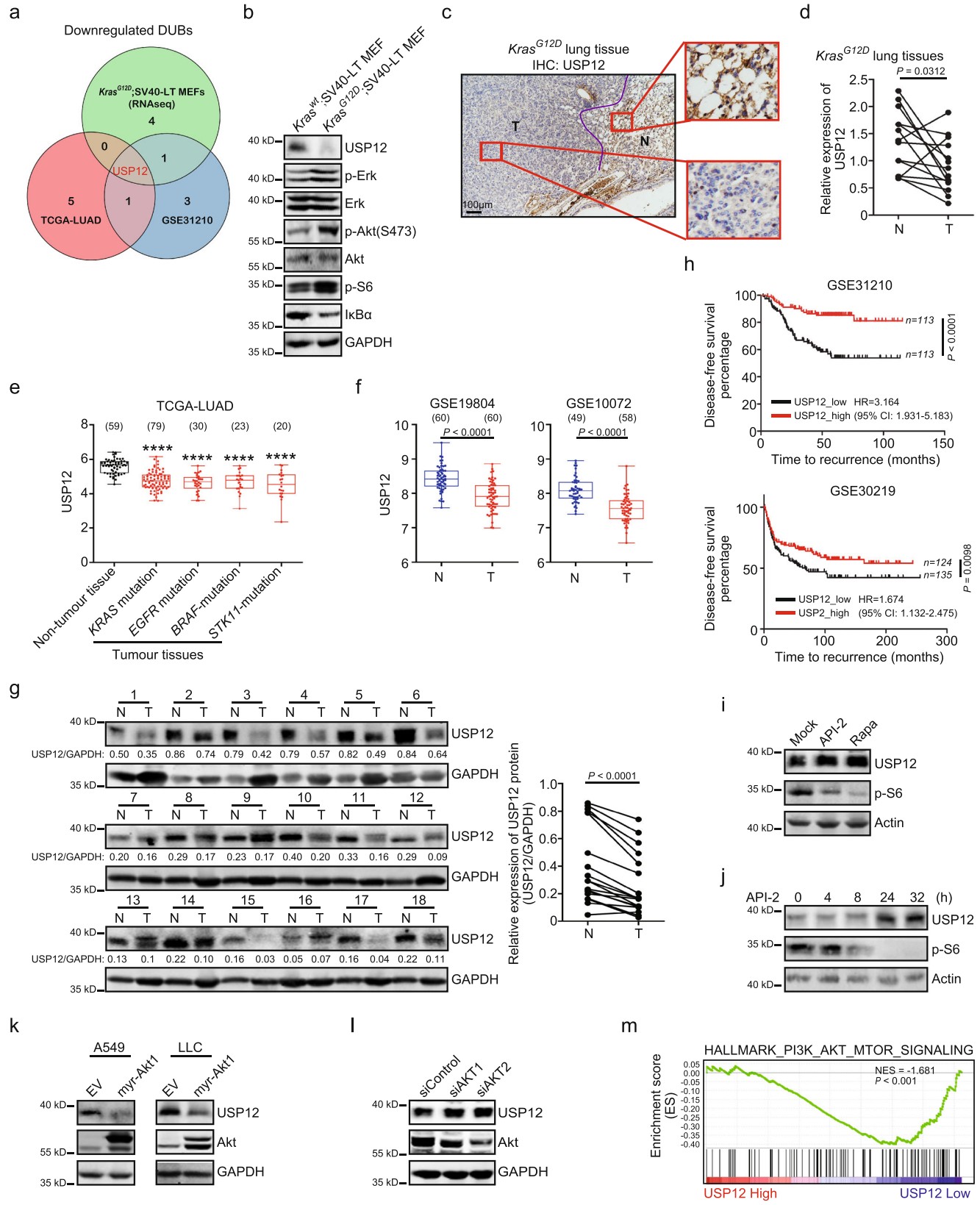

responsible for the downregulation of USP12 by AKT-mTOR signalling (Supplementary Fig. 1h). Furthermore, gene set enrichment analysis (GSEA) confirmed a negative correlation between the transcript levels of USP12 and AKT-mTOR

activation in NSCLC (Fig. 1m). We also examined whether a mutual regulation occurred between USP12 and AKT, and the results showed that USP12 overexpression did not significantly affect AKT phosphorylation in human lung cancer cells

**Fig. 1 USP12 expression is commonly downregulated in human NSCLC and *Kras*$^{G12D}$-driven mouse lung tumours. a** Venn diagram showing the overlapped DUB genes that were significantly downregulated in cells with Kras activation and clinical NSCLCs. Data from our RNA-seq analysis of tumourigenic *Kras*$^{G12D}$;SV40-LT MEFs vs. non-tumourigenic *Kras*$^{wt}$;SV40-LT MEFs, and from TCGA and GSE31210. **b** Western blotting showing indicated protein levels in MEFs with or without Kras$^{G12D}$ expression. **c, d** Representative immunohistochemistry (IHC) staining (**c**, representative images of three biologically independent mice) and USP12 mRNA levels (**d**) in lung tissues from *Kras*$^{LSL-G12D/+}$ mice 2 months after Ad-Cre infection. 2-tailed paired *t*-test. T: tumour; N: non-tumour tissue. **e** Boxplot showing USP12 levels in NSCLC patients with the mutations of *KRAS*, *EGFR*, *BRAF* and *STK11* genes. The data were obtained from the TCGA-LUAD database. ****$P < 0.0001$ vs. non-tumour tissue by 2-tailed unpaired *t*-test. **f** Boxplots of USP12 gene expression in the NSCLC samples from the GSE19804 and GSE10072 databases. 2-tailed unpaired *t*-test. **g** USP12 protein levels in human NSCLC samples. The quantification analysis is shown on the right ($n = 18$). 2-tailed paired *t*-test. **h** Kaplan-Meier plots showing the disease-free survival of NSCLC patients based on USP12 expression. Two-sided log-rank test. HR: hazard ratios; CI: confidence interval. **i** Levels of indicated proteins in A549 cells 24 h after API-2 (50 μM) or rapamycin (Rapa; 100 nM) treatment. **j** Levels of USP12, p-S6, and Actin in A549 cells treated with API-2 for indicated periods. **k, l** Protein levels of USP12, Akt, and GAPDH in indicated cells. **m** Gene set enrichment analysis (GSEA) of the GSE31210 database with the PI3K-AKT-MTOR signature and USP12 transcript levels. NES: Normalized Enrichment Scores. Sample sizes for each group are given in parentheses (**e**, **f**). For boxplots, the centre mark represents the median, and whiskers show minimum/maximum values.

(Supplementary Fig. 1i, j). Collectively, these data indicate that USP12 downregulation is causally associated with oncogenic activation of AKT-mTOR signalling in NSCLC.

**USP12 suppresses lung tumour growth.** Given the abundant USP12 downregulation in NSCLC and mouse lung tumours, we next examined whether USP12 regulated tumour growth in the *Kras*$^{G12D}$-driven lung adenoma model by lentivirally delivery of Cre recombinase alone or Cre together with USP12 (USP12;Cre). Exogenous USP12 expression markedly increased the survival of mice and decreased overall tumour number or burden in the entire lung compared with that of control mice (Fig. 2a–c). To corroborate these observations, we compared the tumour growth of Lewis lung carcinoma (LLC) cells with or without genetic manipulation of USP12 expression. Tumour growth was significantly enhanced in cells with USP12 shRNA expression compared with control cells (Fig. 2d). Next, we used a single-cell culture system to generate isogenic clones of LLC cells with high or low levels of USP12 to further elucidate the functional relevance of USP12 expression (Fig. 2e). Among a panel of sub-cell lines generated, 8 clones with high or low expression of USP12 (USP12$^{high}$ or USP12$^{low}$) were used for subcutaneous implantation in mice. Notably, compared with USP12$^{low}$ clones, USP12$^{high}$ clones displayed low tumourigenic abilities or slow tumour growth rates in mice (Fig. 2f). Together, the results from genetically engineered mouse and transplanted tumour models suggest that USP12 negatively regulates lung tumour growth.

**USP12 negatively regulates protumourigenic chemokine production.** To explore the molecular mechanism underlying USP12-mediated tumour suppression, we analysed the transcriptional profiles of lung tumour cells with or without USP12 overexpression by RNA-seq. Gene ontology (GO) analysis revealed that the pathways significantly enriched in USP12-downregulated genes were mostly related to cytokine-mediated signalling pathway and immune effector process (Fig. 3a); further in-depth examination showed that USP12 expression decreased levels of a number of chemokine, such as CXCL8, CXCL1, CXCL2, CCL2 and CCL5, most of which were associated with immune cell recruitment (Fig. 3b and Supplementary Fig. 2a).

To further validate these findings, we measured a panel of chemokine in a conditioned medium (CM) derived from A549 cells with or without ectopic USP12 expression using protein arrays. Consistently, the most significantly affected chemokine by USP12 included CXCL8, CXCL1, CCL2 and CCL5, whereas CXCL2 was not detectable in the CM of A549 cells (Fig. 3c). Small hairpin RNA-mediated silencing of USP12 confirmed the findings in both human and mouse lung cancer cells, as evidenced

by ELISA quantification (Supplementary Fig. 2b, c). Moreover, decreased levels of these chemokines were only observed in the supernatants of tumour cells expressing USP12-WT but not in those of tumour cells with USP12-C48S (Fig. 3d and Supplementary Fig. 2d), a catalytically inactive USP12 mutant[16]; a similar phenomenon was observed in LLC tumour tissues expressing the different forms of USP12 (Supplementary Fig. 2e). Of note, the USP12-mediated regulation was also confirmed in single-cell LLC clones by measuring the mRNA expression of Usp12, Cxcl1 and Ccl2 (Supplementary Fig. 2f). Given that USP12 is negatively regulated by AKT signalling (Fig. 1i–m), we then tested whether restoring the expression of USP12 could counterbalance the activity of AKT in the regulation of chemokine expression. The results showed that the myr-AKT1-induced upregulation of CXCL8 and CXCL1 expression was abrogated upon USP12 overexpression (Fig. 3e). Furthermore, overexpression of USP12 but not of USP12-C48S inhibited LLC and 889-S1 tumour growth in immune-competent mice (Supplementary Fig. 2g, h), indicating that the deubiquitinase activity was required for USP12 to exert its tumour suppressive function. More importantly, the inhibitory effects of USP12 on tumour growth were not observed in immune-deficient mice (Supplementary Fig. 2g, h).

To further detect whether the alterations in the chemokine profiles were critical in USP12-mediated control of tumour growth, we overexpressed Cxcl1 combined with or without Ccl2 into mouse lung tumour cells and found that LLC cells, when overexpressing Cxcl1, grew more rapidly than control cells (Fig. 3f). Similarly, co-expression of Cxcl1 and Ccl2 in 889-S1 tumour cells substantially increased tumour incidence and growth in the presence of USP12 overexpression (Fig. 3g), suggesting the importance of the tumour-extrinsic effects controlled by USP12. Taken together, these data suggest that downregulation of USP12 in NSCLC may help establish a tumour microenvironment that is favourable for tumour growth by facilitating protumourigenic chemokine expression.

**USP12 regulates chemokine production by deubiquitinating PPM1B.** Given the relevance of the USP12-mediated chemokine, such as CXCL8, CXCL1 and CCL2, in tumour development and progression, we examined the mechanism underlying the USP12-mediated regulation. Our transcriptome analysis revealed that a number of genes related to the NF-κB signalling pathway, essential signalling for cytokine expression, were remarkably associated with USP12 expression (Fig. 4a and Supplementary Fig. 3a). Consistently, concurrent downregulation of USP12 and IκB, a cytosolic NF-κB inhibitory protein, was observed in SV40-LT MEFs upon Kras$^{G12D}$ activation (Fig. 1b). NF-κB reporter assays confirmed USP12-mediated suppression of NF-κB

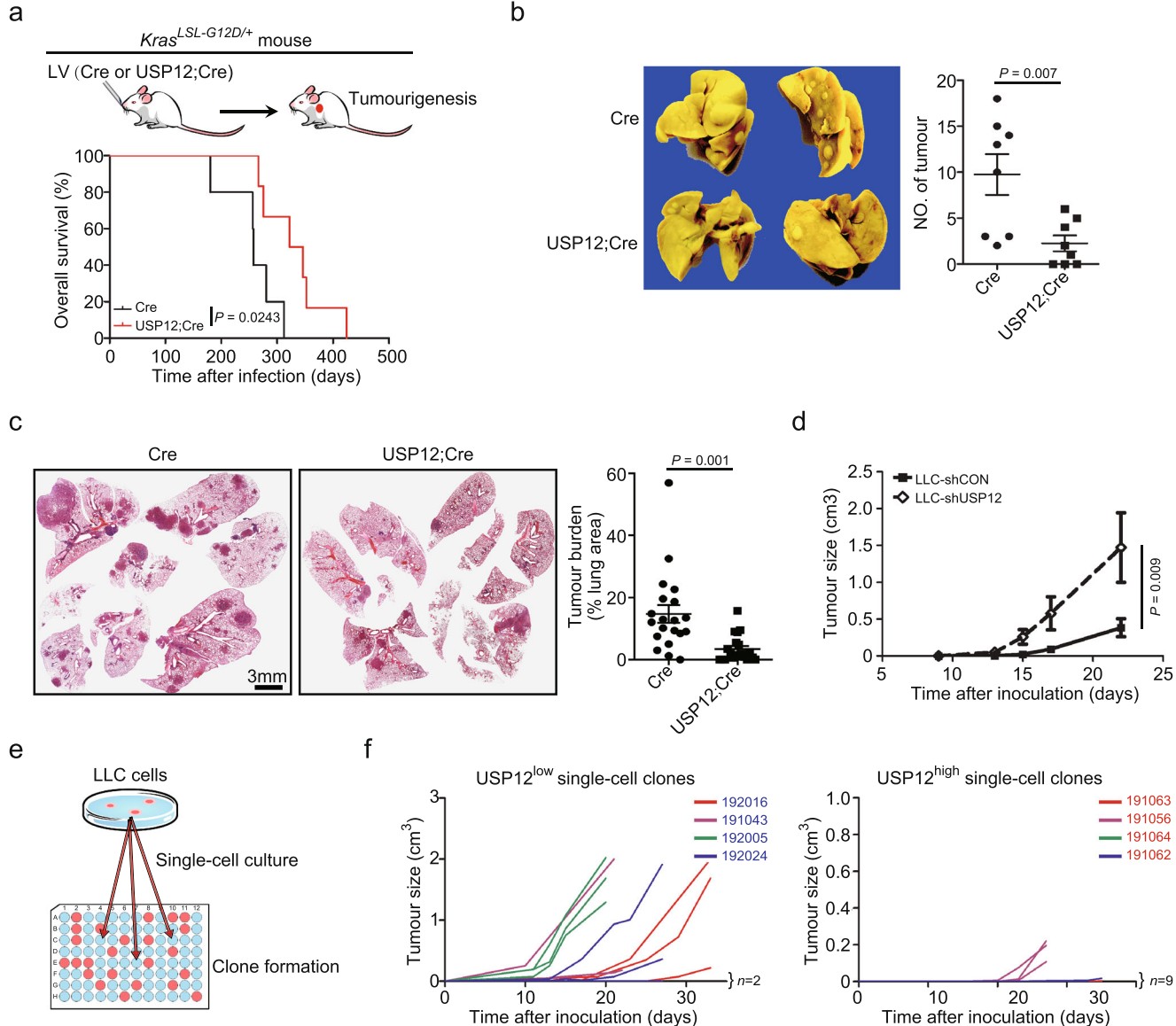

**Fig. 2 USP12 negatively regulates lung tumour growth. a** Schematic diagram showing the intranasal lentiviral delivery of Cre and USP12 plus Cre (USP12;Cre) in Kras$^{G12D/+}$-induced lung tumour (top). LV: lentivirus. Kaplan-Meier plots showing the overall survival of indicated Kras$^{LSL-G12D/+}$ mice ($n = 5-6$) (bottom). Two-sided log-rank test. **b** Representative images of the lungs and statistical analysis of tumour numbers in indicated mouse lungs of Kras$^{LSL-G12D/+}$ mice 8 months after lentiviral infection (mean ± SEM, $n = 8$ each group). 2-tailed unpaired $t$-test. **c** Representative images of H&E-stained lung sections (left) and quantitation of tumour burden in lung lesions described in **b** (mean ± SEM, $n = 20$ lung lesions each group) (right). 2-tailed unpaired $t$-test. **d** Subcutaneous tumour growth of LLC cells stably transduced with control shRNA (LLC-shCON) or shRNA targeting USP12 (LLC-shUSP12) (mean ± SEM, $n = 5$ each group). Two-sided Mann-Whitney U-test to calculate the differences between the tumour sizes of two groups at day 22. **e** Schematic diagram showing generation of single-cell clones from LLC cells. **f** Tumour growth kinetics of LLC single-cell clones with low (USP12$^{low}$) or high (USP12$^{high}$) expression of USP12 ($n = 3-4$ each clone).

signalling activity in tumour cells (Supplementary Fig. 3b, c). These results suggest that USP12 represses NF-κB signalling activity to regulate chemokine production.

Next, we performed immunoprecipitation-mass spectrometry (IP-MS) analysis to find proteins potentially interacting with USP12 (Fig. 4b). Among the proteins identified, the phosphatase PPM1B was chosen for further study since it functions as an NF-κB inhibitor by inactivating IKKβ[22]. The interaction between PPM1B and USP12 was then validated (Fig. 4c), raising the question of whether USP12 deubiquitinates PPM1B. The ubiquitination experiments showed that ectopic expression of USP12 but not the C48S mutant decreased the levels of PPM1B

polyubiquitination (Fig. 4d). Furthermore, forced USP12 expression significantly potentiated the protein stability of PPM1B (Supplementary Fig. 3d), and the knockdown of WDR48, a binding partner of USP12 that was reported to contribute to USP12 deubiquitinase activity[16], suppressed PPM1B expression (Supplementary Fig. 3e). As expected, USP12 knockdown decreased the levels of both PPM1B and IκB (Fig. 4e). Consistent with these observations, PPM1B significantly inhibited basal or shUSP12-upregulated NF-κB activity in tumour cells (Fig. 4f), and the USP12-mediated inhibition of chemokine expression was counteracted by PPM1B silencing (Fig. 4g). Importantly, ectopic PPM1B expression inhibited tumour growth under steady-state

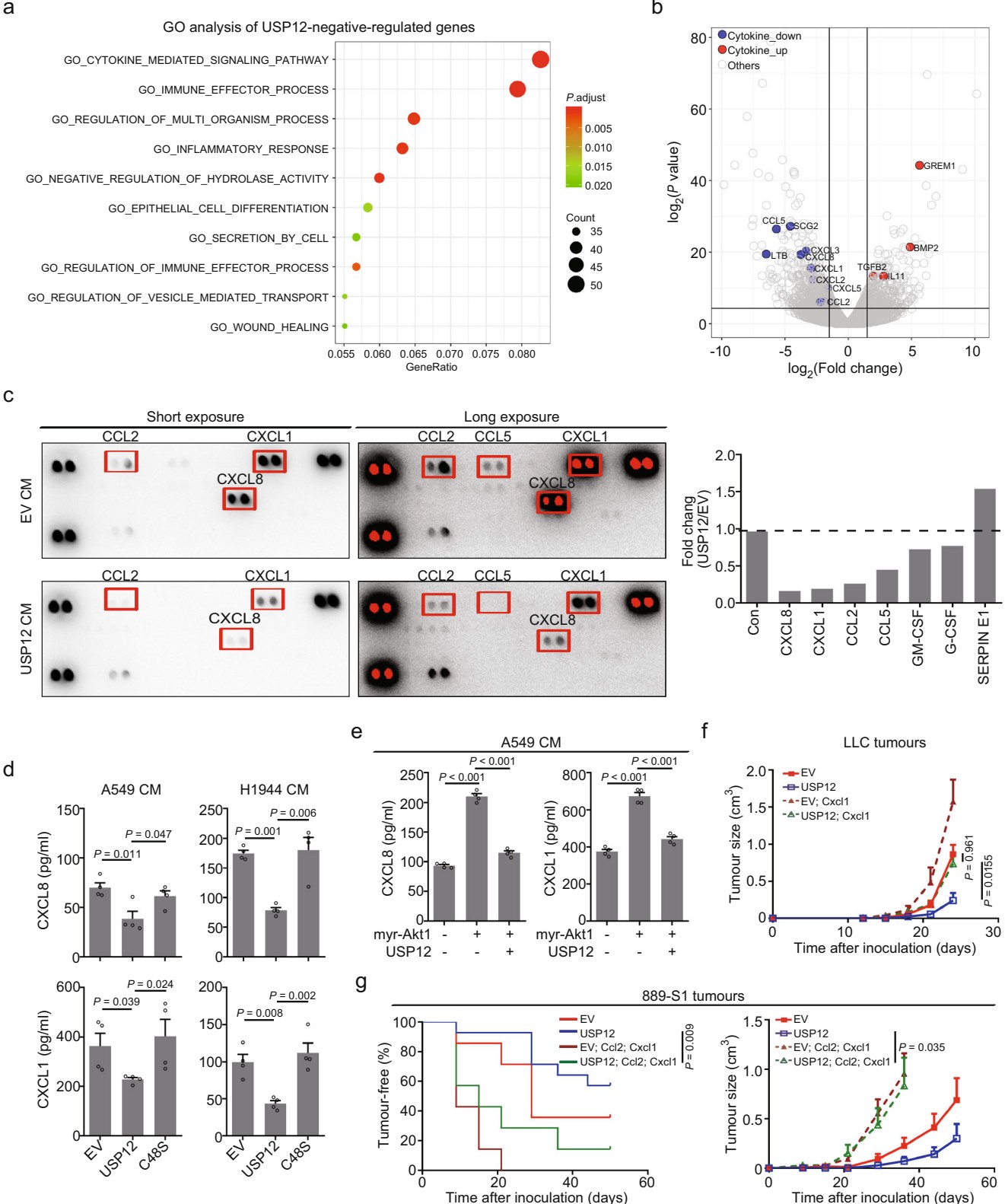

conditions or in the accelerated course caused by USP12 knockdown (Fig. 4h). The regulation of PPM1B by USP12 was recapitulated in mouse $Kras^{G12D}$-driven lung adenomas and some of the human NSCLC tissues examined, as evidenced by low levels of PPM1B and USP12 concurrently presented in tumour cells (Fig. 1c and Fig. 4i, j). Taken together, these data suggest that USP12 inhibits lung tumour growth and NF-κB-dependent

chemokine expression by deubiquitinating and stabilizing PPM1B.

USP46 is a DUB with a high degree of sequence similarity to USP12. We then tested whether USP46 could regulate PPM1B expression. We found that USP46 could be co-immunoprecipitated with PPM1B (Supplementary Fig. 3f) and that USP46 over-expression increased levels of PPM1B protein to some extent

**Fig. 3 USP12 regulates chemokine expression in lung tumour cells. a** Gene ontology (GO) analysis of USP12-downregulated genes in A549 cells using compilation C5 (MSigDB). Downregulated genes were determined with the following criteria: $P \leq 0.05$ and $\log_2$(fold change) $\leq -1$. **b** Volcano plot showing the differentially expressed genes affected by USP12 expression. The cytokine genes are highlighted with colour. **c** Chemokine levels in conditioned medium (CM) from A549-EV or A549-USP12 cells measured by protein array (left). The signal intensity was quantified as a fold change in A549-USP12 vs. A549-EV cells (right). EV: vector control. Con: control. **d**, **e** ELISA analysis of CXCL8 and CXCL1 levels in CM of indicated cells (mean ± SEM, $n = 4$ per group). Kruskal-Wallis test. **f** Tumour growth of the indicated LLC cells is shown as the mean ± SEM ($n = 6$ per group). Two-way ANOVA followed by Bonferroni's multiple comparisons post-test. **g** Kaplan-Meier plots showing the tumour-free period (left, Two-sided log-rank test) and tumour growth (right, Two-way ANOVA followed by Bonferroni's multiple comparisons post-test) of indicated 889-S1 cells ($n = 7-14$ per group).

(Supplementary Fig. 3g), whereas silencing USP46 resulted in a slight decrease in PPM1B expression (Supplementary Fig. 3h). Although these results indicate that USP46 may regulate PPM1B expression in cultured cells, clinical data from the transcriptional datasets of NSCLC revealed that USP12 but not USP46 expression was downregulated in tumour specimens compared with normal tissues (Supplementary Fig. 3i and Fig. 1e, f) and that the abundance of USP46 transcripts in tumours was apparently lower in comparison with USP12 mRNA levels (Supplementary Fig. 3j). These results suggest that downregulation of USP12 accounts for the impairment in PPM1B expression in human NSCLC samples.

**USP12 downregulation promotes the development of immunosuppressive TME.** Given that USP12 negatively regulated protumourigenic chemokine expression and exerted a tumour-suppressive effect in immune-competent mice but not in immune-deficient mice, it seems intuitive to speculate that USP12 downregulation in tumours may help establish a favourable microenvironment for tumour growth. FACS analyses of immune cell profiles (Supplementary Fig. 4) revealed an increase in the percentage of infiltrating tumour-associated macrophages (TAMs) and in their expression of PD-L1, as well as in the fraction of CD4$^+$Foxp3$^+$CD25$^+$ Treg cells in subcutaneous mouse LLC tumours with USP12 silencing, whereas trends towards increased the frequencies of myeloid-derived suppressor cells (MDSCs) and decreased proportion of T cells in these tumours were also observed, although the differences did not reach statistical significance (Fig. 5a, b and Supplementary Fig. 5). Intriguingly, enhanced PD-L1 expression was also detected in CD45$^-$ cells in shUSP12 tumours (Fig. 5c), indicating a role of USP12 in modulating this immunoregulatory molecule in tumour cells and non-hematopoietic stromal cells in vivo. Since the chemokine affected by USP12, such as CXCL8 and CXCL1, and the other macrophage-related factors, are involved in tumour angiogenesis[23], we evaluated the proportion of CD31$^+$ cells in LLC tumours, and observed an increase in the subpopulation in tumours with USP12-knockdown (Fig. 5d). Accordingly, in mouse Kras$^{G12D}$-driven lung tumours, decreased TAM numbers and less vascularization was markedly observed in USP12;Cre tumours compared with control tumours (Supplementary Fig. 6a, b), whereas even though no significant change was observed in the proportion of infiltrating CD4$^+$ or CD8$^+$ T cells, T cell activation appeared to be strengthened by USP12 expression, as evidenced by CD44, PD-1, CD69, IFN-γ and TNF-α staining (Supplementary Fig. 6c, d). In vitro macrophage recruitment experiments showed that CM from tumour cells ectopically expressing USP12 but not USP12-C48S induced less migration of bone-marrow-derived macrophages (BMDMs) than the supernatant of control cells, whereas CM from LLC-shUSP12 cells exacerbated the BMDM migration (Supplementary Fig. 7a, b). We also investigated whether USP12 expression in tumour cells might modulate macrophage polarization. Compared with CM from control LLC cells, CM from shUSP12 cells induced higher levels of the M2-like macrophage markers Il10 and Ym1, which

coincided with lower levels of the M1-like macrophage markers Tnf-α, Mx1, and Il33, in BMDMs (Supplementary Fig. 7c). Consistent with this observation, BMDMs expressing CD206, a marker of alternatively activated M2-like macrophages, were substantially increased due to USP12 inhibition in tumour cells (Supplementary Fig. 7d).

We further examined whether USP12-mediated regulation of TME and angiogenesis was related to its ability to control the expression of the chemokine. It was found that Cxcl1 over-expression in LLC tumours was sufficient to restore TAM infiltration and PD-L1 expression on TAMs, and led to a significant, albeit not complete, the rescue of CD31-positive cell abundance (Supplementary Fig. 7e, f). Similarly, combined expression of Cxcl1 and Ccl2 in 889-S1 tumour cells substantially counteracted the inhibitory effects of USP12 on the TAM infiltration and their PD-L1 expression as well as on CD31$^+$ cell presence (Fig. 5e, f). Moreover, coexpression of Cxcl1 and Ccl2 efficiently attenuated activation of CD8$^+$ and CD4$^+$ T cells in tumour cells ectopically expressing USP12, as evidenced by the changes in the frequencies of TNF-α$^+$IFN-γ$^+$ cells in the populations (Fig. 5g). Collectively, these data indicate that dysregulated chemokine expression caused by USP12 down-regulation, as exemplified by CXCL1 and CCL2, is important for the development of immune-suppressive TME.

To validate whether the role of USP12 on TME reprogramming has pathological relevance in human NSCLC, we investigated the relationship between TAMs and USP12 expression in human NSCLC cohorts. The NSCLC samples were first scored according to the macrophage signature using the method xCell[24], and then macrophage enrichment in patients with high or low USP12 levels was estimated. The scores associated with the presence of macrophages in USP12_high patients were lower than those in USP12_low patients (Fig. 5h), supporting a negative relationship between USP12 expression and macrophage accumulation in NSCLC patients. In addition, enrichment score analysis of the transcript levels of myeloid cell markers and cytokines[25] demonstrated that low USP12 expression was associated with the enrichment of myeloid cell markers and cytokines (Fig. 5i). Among them, CD163, a biomarker of M2 macrophages, was correlated with poor outcomes in NSCLC[26,27]. The significant inverse relationship between USP12 and CD163 was also demonstrated in other NSCLC databases (Supplementary Fig. 7g). Moreover, a negative correlation between CD274 (PD-L1) and USP12 transcript levels were found in NSCLC tumours (Supplementary Fig. 7g). GSEA validated an inverse correlation between USP12 expression and tumour angiogenesis in NSCLC (Fig. 5j). Furthermore, we used recently developed computational methods, Tumour Immune Dysfunction and Exclusion (TIDE)[28], to measure the level of T cell dysfunction and T cell exclusion in NSCLC patients. Accordingly, the TIDE scores in USP12_low patients were higher than those in USP12_high patients (Supplementary Fig. 7h). Taken together, these results indicate that USP12 expression in tumour cells plays an important role in reprogramming the TME.

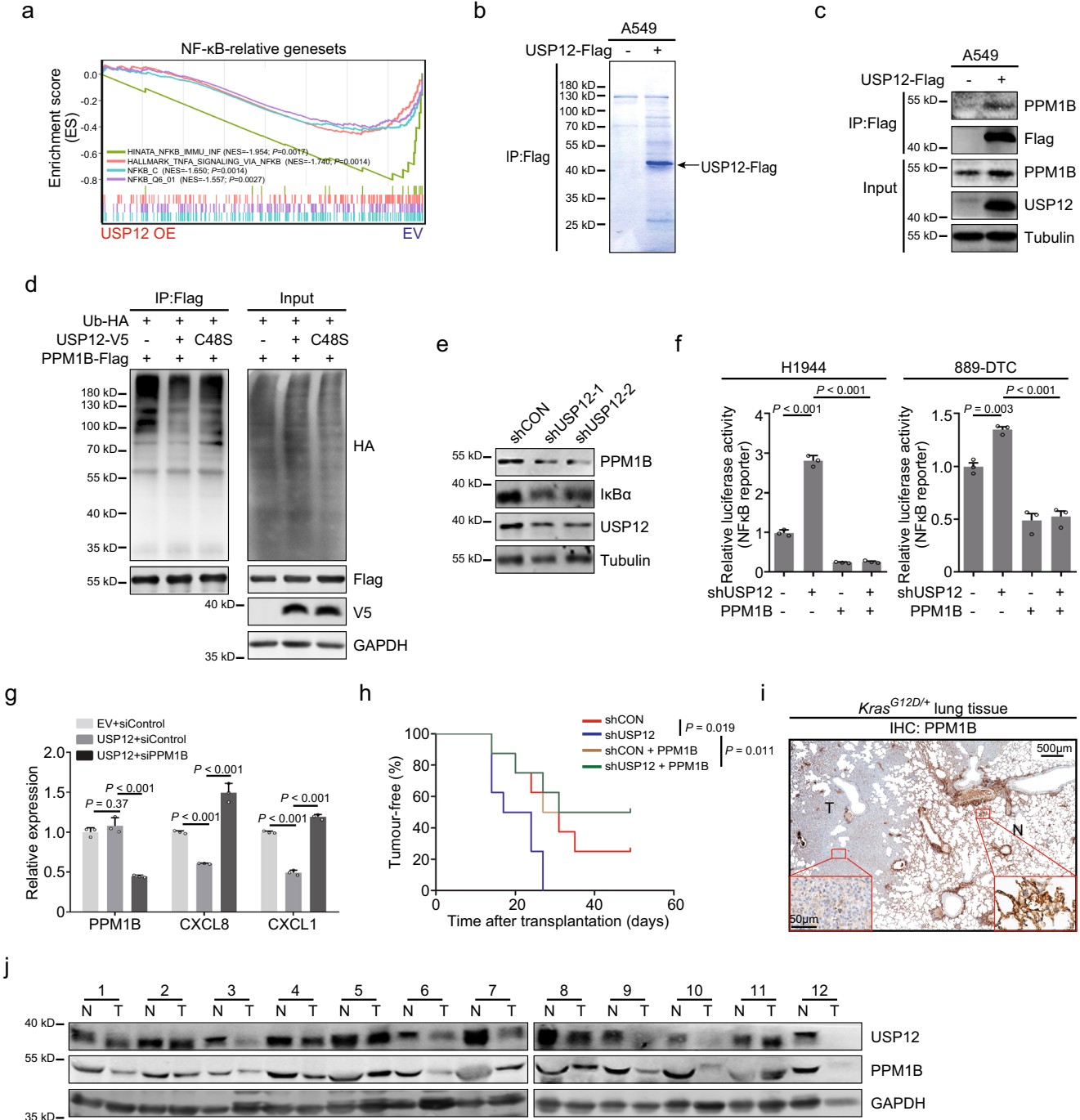

**Fig. 4 USP12 deubiquitinates and stabilizes PPM1B. a** GSEA of the transcriptional profiles of USP12-overexpressed (OE) A549 cells and control cells (EV) with the NF-κB-related signatures. **b** The immunoprecipitates (IP) captured by an anti-Flag antibody from indicated cells were separated by SDS-PAGE gel and analysed by mass spectrometry. **c** Cell lysates with or without USP12-Flag were immunoprecipitated, and endogenous PPM1B was examined by immunoblotting. **d** PPM1B-Flag and ubiquitin-HA were co-expressed with wide type (WT) USP12 or the catalytically inactive mutant (C48S) in 293T cells. PPM1B was subjected to IP, and the polyubiquitination of PPM1B was assessed by immunoblotting using an anti-HA antibody. **e** 889-DTC cells with USP12 silencing (shUSP12) were immunoblotted with indicated antibodies. **f** The relative luciferase activities were analysed in the indicated cells (mean ± SEM, $n = 3$ per group). One-way ANOVA followed by Tukey's HSD test. **g** Quantification of PPM1B, CXCL1 and CXCL8 mRNA levels in indicated H358 cells (mean ± SEM, $n = 3$ per group). One-way ANOVA followed by Tukey's HSD test. **h** Kaplan-Meier plots showing the tumour-free period after transplantation with indicated 889-S1 cells ($n = 8$ per group). Two-sided log-rank test. **i** Representative IHC staining of PPM1B in lung tissues from $Kras^{G12D/+}$ mice. Representative images are from three biologically independent mice. **j** Protein levels of USP12 and PPM1B in human NSCLC samples.

**USP12 inhibition desensitizes the tumour response to anti-PD-1 therapy.** Based on the findings that USP12 downregulation resulted in increased TAM recruitment, enhanced PD-L1 expression and impaired T cell activation in syngeneic mouse tumours, we, therefore, extended our study to observe whether USP12 in cancer cells could influence their response to PD-1 immune checkpoint blockade. Using 889-S1 cells that express high levels of MHC I upon IFN-γ stimulation and display more aggressive tumour growth in

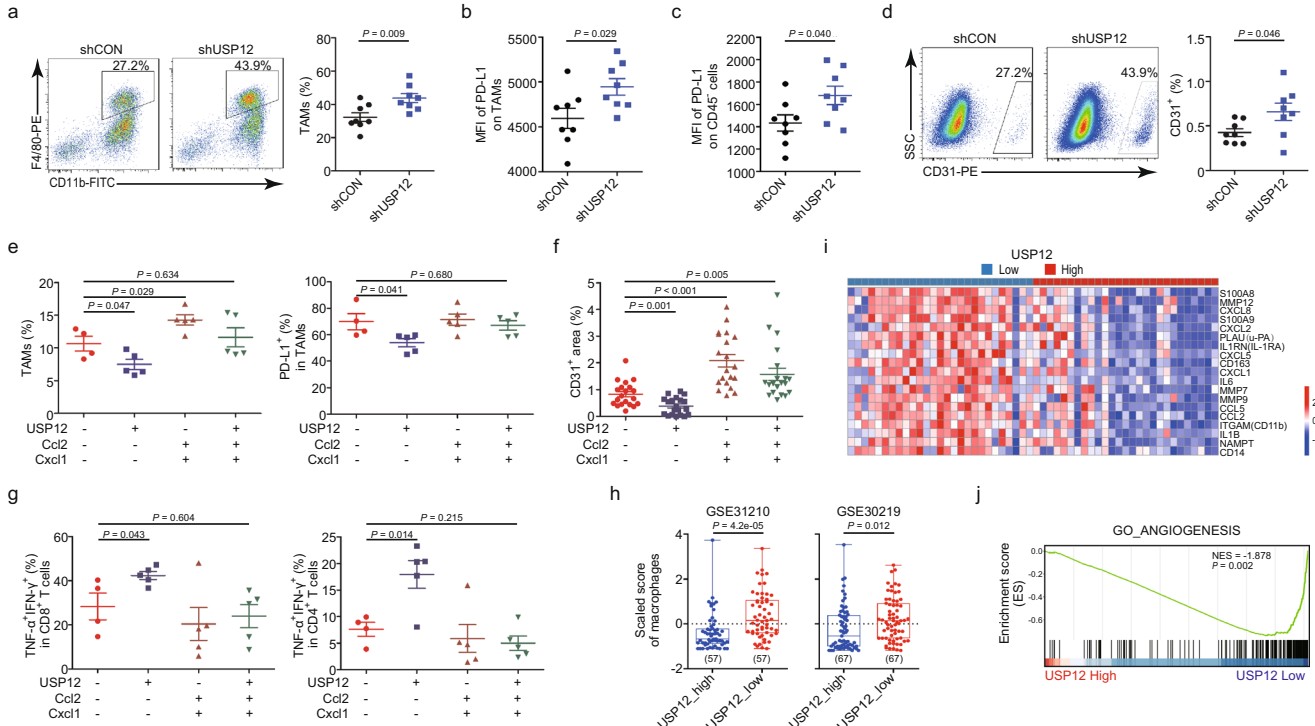

**Fig. 5 USP12 modulates immune cell composition and activation in tumour microenvironment. a–d** Flow cytometric analysis of the proportion of TAMs (**a**), PD-L1 expression on TAMs (**b**), PD-L1 expression on CD45− cells (**c**), and the proportion of CD31+CD45− cells (**d**) in LLC tumours (mean ± SEM, $n = 8$ per group). 2-tailed unpaired $t$-test. **e** Flow cytometric analysis of the proportion of TAMs and PD-L1 expression on TAMs in indicated 889-S1 tumours (mean ± SEM, $n = 4−5$ per group). One-way ANOVA followed by Tukey's HSD test. **f** Quantification analysis of CD31 immunostaining of indicated 889-S1 tumours (mean ± SEM, $n = 20$ fields of view from >3 mice). Kruskal-Wallis test. **g** The proportion of TNF-α+IFN-γ+ cells gated on CD8 and CD4 T cells in indicated 889-S1 tumours (mean ± SEM, $n = 4$-5 per group). One-way ANOVA followed by Tukey's HSD test. **h** Boxplot showing the xCell scores for macrophages in USP12_low (bottom 25%) and USP12_high (top 25%) NSCLC samples. The centre mark represents the median, and whiskers show minimum/maximum values. Sample sizes for each group are given in parentheses. 2-tailed unpaired $t$-test. **i** Heat map of M2 myeloid cell markers and cytokines that were negatively correlated with USP12 ($P < 0.05$), showing 20% of samples from the GSE30219 database with the highest or lowest USP12 expression. **j** GSEA of the GSE30219 database with the angiogenesis signature. Based on the expression of USP12, samples in the top 1/4 were designated high, and those in the bottom 1/4 were designated low.

nude mice than in C57BL/6 mice, we tested the impact of USP12 inhibition on the efficacy of anti-PD-1 therapy. As expected, 889-S1 tumours were sensitive to PD-1 blockade treatment; USP12 silencing, however, substantially desensitized tumours to anti-PD-1 therapy, as evidenced by tumour incidence and volume (Fig. 6a). Furthermore, we analysed the gene expression data of NSCLC tumours stratified by levels of USP12 transcripts with the TIDE model, which could be used to predict checkpoint blockade (ICB) response in different types of cancer-based on integration analysing expression signatures of T-cell dysfunction and T-cell exclusion[28]. The results revealed that the percentage of predicted ICB responders based on the TIDE score in the USP12_low group was significantly lower than that in the USP12_high group (Fig. 6b). Taken together, these results suggest that USP12 in lung cancer cells may function to regulate the tumour response to anti-PD-1 therapy.

## Discussion

Oncogenic mutations drive cancer development through regulating both cancer cell-intrinsic and cancer cell-extrinsic events, which collectively support the development of an immunosuppressive TME[1]. Lung cancer, especially NSCLC, has particularly high somatic mutations, including mutations in *KRAS*, *EGFR* and other surrogates, which share similar outputs at the downstream end of their signalling pathways[11,29]. Here, we identified USP12 as a deubiquitinase broadly downregulated in NSCLC and showed that

convergent AKT-mTOR activation downstream of the oncogenic mutations was responsible for the inhibited expression of USP12. We found that USP12 downregulation caused insufficient deubiquitination of PPM1B and thereby facilitated tumour growth by promoting the generation of the protumourigenic milieu. Finally, we suggest that USP12 downregulation in tumours may account for the impaired response to anti-PD-1 treatment in a mouse tumour model. These findings reveal a molecular control of the development of the tumour-promoting microenvironment in NSCLC and support the potential of manipulating USP12-PPM1B signalling in cancer immunotherapy.

Imbalanced processes of protein ubiquitination and deubiquitination contribute to tumour development and progression[30–32]; aberrations in certain deubiquitinases, mainly due to altered expression or inactivating mutations, are often observed in various types of cancer[13,15,33]. Prior studies suggested a role of USP12 in prostate carcinomas, wherein USP12 promoted AR signalling and tumour cell proliferation, as well as therapy resistance, by regulating AR ubiquitination and MDM2-P53 signalling[19,20,34]. However, USP12 has also been shown to inhibit AKT-dependent cell survival by stabilizing the pro-apoptotic phosphatases PHLPP1 and/or PHLPPL[16,34]. The pathologic function of USP12 in other types of cancer, especially in terms of TME regulation, remains elusive. Thought to act as a downstream mediator of numerous oncogenic drivers, such as *KRAS*, *EGFR*, or mutated *PTEN*, PI3K-AKT-mTOR pathway signalling controls tumour development and

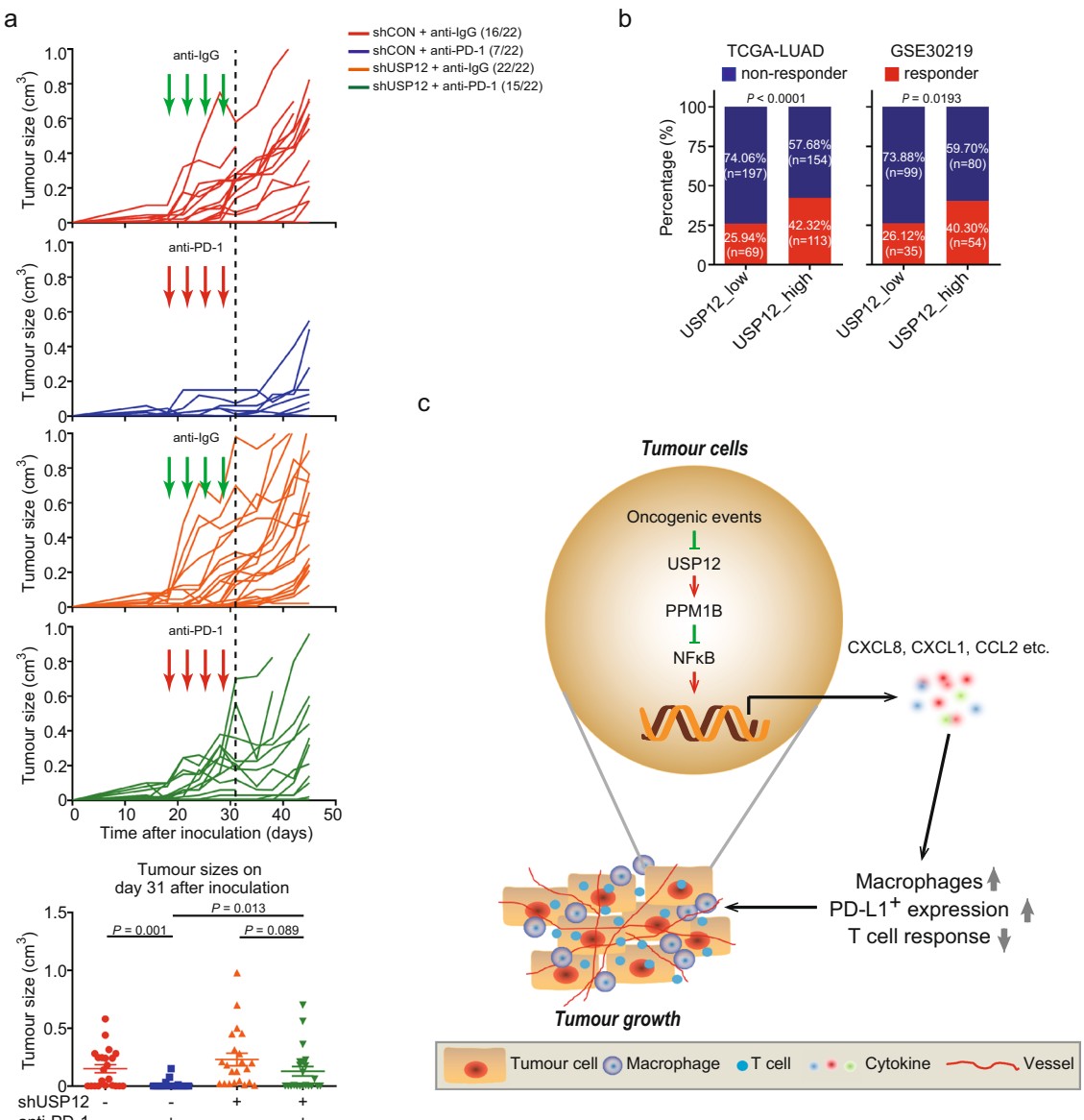

**Fig. 6 USP12 inhibition renders lung cancer cells resistant to anti-PD-1 immunotherapy. a** Top, growth kinetics in shCON- or shUSP12- 889-S1 tumours treated with anti-PD-1 or anti-IgG at days 18, 21, 24, and 28 ($n = 22$ tumours from 11 mice). Bottom, tumour sizes of individual mice at day 31 (mean ± SEM, $n = 22$ per group). Kruskal-Wallis test. **b** Percentage of predicted responder and non-responder patients in the USP12_low and USP12_high groups from the TCGA-LUAD or GSE30219 databases. Fisher's Exact Test. **c** Proposed model of the suppressive function of USP12 in the NSCLC development. USP12 downregulation by oncogenic events in NSCLC impairs PPM1B restriction of NF-κB activation and thereby increases the expression of protumourigenic chemokine, leading to accelerated macrophage infiltration, angiogenesis, and T cell inactivation, and facilitating tumour growth and resistance to ICB treatment.

regulates the cellular composition of the TME[29,35,36]. The loss of PTEN leads to aggressive expression of IL-6, IL-10 and VEGF, thereby facilitating the immune response against cancer cells[37]. A lack of PTEN and activation of the PI3K-AKT pathway yield decreased T cell recruitment and function through VEGF production[6]. This study demonstrated that AKT-mTOR hyper-activation resulted in USP12 downregulation, which in turn increased the production of protumourigenic chemokine in NSCLC, indicating that USP12 is a regulatory node in oncogenic mutation-driven TME development.

The tumour suppressive function conveyed by USP12 is virtually related to the control of protumourigenic chemokine expression. Among the chemokines regulated by USP12 in tumour cells, CXCL1 and CXCL8 share the chemokine receptor CXCR2 and are strong chemoattractants that recruit TAMs and

MDSCs[38–40]. Moreover, CXCL1 and CXCL8 are involved in the angiogenesis process by stimulating endothelial cells or indirectly by modulating myeloid cells to secret proangiogenic factors and MMPs[4,23]. CCL2, another chemokine downregulated by USP12, also contributes to the recruitment and polarization of cancer-promoting myeloid cells[41,42]. Importantly, the inhibitory effects on tumour growth and the alterations in TME composition caused by USP12 overexpression were efficiently rescued by CXCL1 and CCL2. Of note, the chemokine involved in the regulation of TME by USP12 includes, but is apparently not limited to, CXCL1 and CCL2, when considering a collection of the chemokines affected by USP12. Nevertheless, the results of the rescuing experiments exemplified by enforced restoration of these two factors underscore the extrinsic function of USP12 in tumour cells in the regulation of tumour growth. Mechanistically, our

findings identified that PPM1B, a phosphatase of IKKβ[22], is a substrate of USP12. USP12 downregulation in tumour cells accelerates PPM1B ubiquitination and degradation and therefore promotes NF-κB activity in orchestrating the TME. The inter-actions between the PI3K-AKT-mTOR and NF-κB pathways have been highlighted in many studies[36,43,44]. In $Kras^{G12D}$-driven pancreatic tumours, PI3K signalling strongly activates the NF-κB pathway to increase the cytokine network, resulting in increased MDSCs and Tregs surrounding tumour tissues[36]. This study revealed the downregulation of USP12 as a critical event that coordinates AKT-mTOR activation and NF-κB signalling-dependent production of protumourigenic chemokine in NSCLC.

Immunotherapies using anti-PD-L1 and anti-PD-1 antibodies are currently used to treat NSCLC patients. However, the response rates to anti-PD-1 therapy in advanced NSCLC are only 14–20%[45,46]. The loss of PTEN or activation of the PI3K pathway in cancer cells improves resistance to immunotherapy[6,9]. The combination of PI3K-AKT-mTOR inhibitors with anti-PD-1 antibodies may benefit NSCLC patients[29]. Our results provide evidence that USP12 downregulation, which is at least partially due to PI3K-AKT-mTOR hyperactivation, functionally induces tumour cell resistance to PD-1 blockade in mice. Interestingly, the PI3K-AKT-mTOR pathway can increase PD-L1 expression in tumour cells[9,29]. Accordingly, we found that silencing USP12 enhanced PD-L1 expression in tumour cells. Bioinformatics analyses of human NSCLC databases predicted that NSCLC patients with high USP12 expression in tumours would be more responsive to ICB therapy. Therefore, USP12 downregulation in tumour cells may serve as a mechanism underlying AKT-mTOR pathway-related resistance to immunotherapy. Further retro-prospective human studies on the association of USP12 expression and ICB response are warranted in the future.

The study demonstrated that the increase in TAM abundance in TME was the most prominent alteration caused by USP12 downregulation in NSCLC. Macrophages in tumour tissues are the main immunosuppressive cellular component and repress T cell-mediated and innate immune cell-mediated control of tumour development[47,48]. Macrophages express inhibitory receptor ligands (PD-L1, B7-1 and B7-2) to suppress the cytotoxic function of T cells and death receptor ligands (FASL and TRAIL) to trigger T cell death, or secrete an array of effectors to suppress T cell effector functions (IL-10 and TGF-β) and recruit Treg cells (CCL5, CCL20, and CCL22)[47]. TAMs are able to exert prolonged interactions with CD8 T cells to inhibit their migration and infiltration into tumour islets, thereby rendering tumours more resistant to anti-PD-1 treatment[49]. Presumably, the increased macrophages might also attenuate the tumour response to anti-PD-1 therapy by inhibiting T cell function or T cell migration. Emerging studies have revealed that a high level of systemic and tumour-associated CXCL8 is associated with reduced responses to immune checkpoint inhibitors in several types of cancer, including NSCLC[50,51]. In this regard, increased CXCL8 produc-tion contributed from increased TAM presence and direct acti-vation of CXCL8 transcription in tumours with USP12 downregulation may greatly contribute to resistant phenotype to PD-1 blockade. Of note, the altered production of the chemokines caused by USP12 downregulation in tumour cells may also reg-ulate T cell infiltration and function. For instance, previous stu-dies have revealed a suppressive function of CCL2 and CCL5 on T cell activity and an inhibitory effect of CXCL1 on T cell infil-tration into tumours[52–54]. Therefore, the combinatory effects of the aberrantly expressed chemokines on growth and ICB response of tumours with USP12 downregulation are possibly fulfilled through regulating different types of immune cells.

Collectively, our study used several models to underscore the importance of USP12 downregulation in creating a TME that promotes NSCLC development and potentially contributes to tumour resistance to immunotherapy (Fig. 6c). Thus, targeting the USP12-PPM1B cascade may perturb the tumour micro-environment and increase the efficacy of ICB therapy for certain cancers.

## Methods

**Cell culture, vectors, and small interfering RNAs.** The human lung adeno-carcinoma cell lines A549, NCI-H358 (H358) and NCI-H1944 (H1944), mouse Lewis lung carcinoma cell line LLC, and HEK293T cell line were purchased from the American Type Culture Collection (ATCC, Manassas VA, USA). The mouse lung cancer cell line 889-DTC was derived from $Kras^{LSL-G12D/+};Trp53^{flox/flox};Rosa26^{LSL-Tomato}$ mice[55]. 889-S1, showing higher tumourigenic ability than 889-DTC, is a single-cell clone of 889-DTC cells, which was generated from 889-DTC tumours in C57BL/6 mice. $Kras^{wt}$ and $Kras^{G12D}$ MEFs were isolated from control and $Kras^{LSL-G12D/+}$ mice according to the protocol described in the literature[56], and immortalized with the SV40 large T antigen followed by Ad-Cre transfection. After cells were initially grown, multiple aliquots were cryopreserved. All cell lines were used within 3 months after resuscitation and routinely examined for myco-plasma contamination. Sequences encoding V5-tagged or 3×Flag-tagged human USP12, catalytically inactive USP12 variant (C48S mutant), V5-tagged or 3×Flag-tagged human PPM1B, 3×Flag-tagged human USP46, 3×Flag-tagged mouse Cxcl1 precursor, and 3×Flag-tagged mouse Ccl2 precursor were subcloned into the pLVX vectors (Clontech 632187 or Clonetech 632183) by PCR. The USP12 promoter sequences with serial deletions were cloned into the pGL3-basic vector (E1751, Promega). The primers used were listed in Supplementary Table 1, 2. The plasmids expressing HA-tagged wild-type ubiquitin (Addgene plasmid 17608) were kindly provided by Ted Dawson. Myr-Akt1 was generated by PCR amplification from pT3-myr-AKT-HA (#31789, Addgene, Cambridge MA, USA) and then cloned into pLVX vector. The IKKβ expression vector and NF-κB luciferase reporter plasmid were gifts from Dr. Xiaoren Zhang (Guangzhou medical university, Guangzhou, China). Sequences encoding shRNAs targeting both human and mouse USP12 (Target sequences: 5'-CATCAATTACTCACTGCTTAA-3' and 5'-CAGTTCTTCA AGCACTTTATT-3'), were cloned into pLKO.1 plasmid (Sigma-Aldrich, Missouri, USA). pLKO-shGFP (Target sequence: 5'-GCTCCGTGAACGGCCACGAGT-3') was used as the shControl-expressing vector (shCON). The lentivirus was gener-ated using the psPAX2 (Addgene plasmid, 12260) and pMD2.G (Addgene plasmid, 12259) packaging plasmids. The siRNAs targeting human AKT1, AKT2, PPM1B, USP12, USP46 and WDR48 were synthesized by the Shanghai Genepharma Co. (Shanghai, China). Human PPM1B (5'-CGAGATAACATGAGTATTG-3'). Human AKT1 (5'-GTGCCATGATCTGTATTTA-3'). Human AKT2 (5'-CCATG AAGATCCTGCGGAA-3'). Human USP12 (si-1: 5'-GTCATAATGTTCATCATA A-3'; si-2: 5'-TTCATAGTATGATTTGTCA-3'). Human USP46 (si-1: 5'-CCATGA AACTTACGCAGTA-3'; si-2: 5'-CACTATTTCGGATTGGTCA-3'; si-3: 5'-GCTT ACCAATGAAACTCGA-3'). Human WDR48 (si-1: 5'-CGACAGTAAAAGTATG GAA-3'; si-2: 5'-GGATGTGAATACTCTAACA-3').

**Antibodies and reagents.** Recombinant human TNFα (AF-300-01A) was from PeproTech (Rocky Hill, NJ). API-2 (2151) was from Tocris (Bristol, UK). Cyclo-heximide (CHX, 2112S), Rapamycin (9904) and U0126 (9903) were from CST (Beverly, MA). Erk inhibitor (328006), WP1066 (573097), c-Myc inhibitor (475946), JNK inhibitor II (420119), SB203580 (559395), SB431542 (616461) and SIS3 (566405) were from Sigma-Aldrich (St Louis, MO). The following primary antibodies are used in this study: USP12 (WB, 1:300; IHC, 1:30; PA568439, Thermo Fisher Scientific), PPM1B (WB, 1:1000; IHC, 1:100; ab137811, Abcam), IkBα (WB, 1:1000; 10268-1-AP, Proteintech), WDR48 (WB, 1:1000; A6854, ABclonal), Phospho-IKKα/β (Ser176/180) (WB, 1:1000; 2697, CST), IKKα (WB, 1:1000; 11930, CST), Phospho-Akt (Ser473) (WB, 1:1000; 4060, CST), Akt (pan) (WB, 1:1000; 2920, CST), Phospho-S6 (Ser235/236) (WB, 1:1000; 4858, CST), Phospho-p44/42 MAPK (Erk1/2) (Thr202/204) (WB, 1:1000; 4370, CST), p44/42 MAPK (Erk1/2) (WB, 1:1000; 4695, CST), Flag-tag (WB, 1:1000; IP, 1:200; F1804, Sigma-Aldrich), V5-tag (WB, 1:1000; IP, 1:200; PM003, MBL), His-tag (WB, 1:1000; IP, 1:200; PM032, MBL), GAPDH (WB, 1:1000; sc-32233, Santa Cruz), β-actin (WB, 1:2000; sc-47778, Santa Cruz), α-Tubulin (WB, 1:2000; sc-69969, Santa Cruz), and CD31 (IHC, 1:100; GB13063, Servicebio). Second antibodies: anti-rabbit IgG HRP-linked antibody (1:10,000; sc-2004, Santa Cruz), anti-mouse IgG HRP-linked antibody (1:10,000; sc-2005, Santa Cruz), anti-rabbit IRDye800CW (1:10,000; 926-32211, Li-COR), and anti-mouse IRDye800CW (1:10,000; 926-32210, Li-COR). The fluorochrome-conjugated antibodies were used for FACS as follows: CD45-PerCP-Cy5.5 (1:250; 45-0451-82, eBioscience), CD45-FITC (1:250; 103108, BioLegend), CD11b-PE-Cy7 (1:250; 25-0112-82, eBioscience), CD11b-FITC (1:250; 11-0112-82, eBioscience), F4/80-PE (1:250; 12-4801-82, eBioscience), CD11c-FITC (1:250; 11-0114-82, eBioscience), Gr-1-APC (1:250; 12-5931-82, eBioscience), MHC II-PE-Cy5 (1:250; 15-5321-81, eBioscience), Ly6G-PE (1:250; 551461, BD Biosciences), PD-L1-PE-Cy7 (1:250; 25-5982-82, eBioscience), CD31-PE (1:250; 561073, BD Biosciences), CD3-FITC (1:250; 100204, BioLegend), CD3-PE (1:250; 12-0031-82, eBioscience), CD8-PE-Cy5 (1:250; 15-0081-82, eBioscience), CD8-PE-Cy7 (1:250; 25-0081-82, eBioscience), CD4-APC (1:250; 17-0041-82, eBioscience), CD4-PE-Cy5 (1:250; 15-0041-82, eBioscience), PD-1-PE

(1:250; 12-9985-82, eBioscience), CD44-FITC (1:250; 11-0441-81, eBioscience), CD69-FITC (1:250; 11-0691-82, eBioscience), Foxp3-PE (1:250; 12-5773-82, eBioscience), CD25-APC (1:250; 101910, BioLegend), IFN-γ-APC (1:250; 17-7311-82, eBioscience), TNF-α-PE (1:250; 12-7321-81, eBioscience), NK1.1-APC (1:250; 17-5941-82, eBioscience), B220-PE-Cy7 (1:250; 25-0452-82, eBioscience), and MHC I-FITC (1:250; 114605, Biolegend).

**Lentivirus production.** Full length of USP12 cDNA was cloned into pLenti-CMV-NLS-Cre-3×FLAG vector (OBiO Technology (Shanghai) Corp., Ltd), which expresses a Cre protein fused with a nuclear localization signal. The lentiviruses were generated by co-transfection of HEK293T cells with lentiviral packaging plasmids psPAX2 and pMD2.G with Lipofectamine® 2000 (Invitrogen). The medium containing lentivirus was harvested at 48 h and 72 h after transfection with a 0.45 μm filter and concentrated by centrifugation (47,800×$g$ for 2 h at 4 °C).

**Mouse models and therapeutic protocols.** The genetically engineered $Kras^{LSL-G12D/+}$ mice, on C57BL/6 background, have been described previously[57] (Jackson 2001). Adenovirus-Cre recombinase (Ad-Cre) are purchased from OBiO Technology (Shanghai) Corp., Ltd. To induce the lung tumour formation, 8–12-week-old $Kras^{LSL-G12D/+}$ mice were anesthetized and then delivered the lentivirus particles ($2 \times 10^6$ per mouse) or Ad-Cre ($1 \times 10^9$ per mouse) using the intranasal delivery method[58]. Mouse LLC cells ($1 - 2 \times 10^6$) or 889-S1 cells ($5 - 10 \times 10^5$) were subcutaneously injected into the left flank of wild-type C57BL/6 male mice (5–6 weeks age) or B-NDG male mice (NOD-$Prkdc^{scid}$ $IL2rg^{tm1}$/Bcgen, BIOCYTOGEN). Tumour volumes were evaluated twice or thrice a week and calculated using the following formula: (width*width*length)/2. Tumours were left to develop for 3 weeks (LLC), or 7 weeks (889-S1) followed by immune-cell analysis and histology. For the in vivo anti-PD-1 treatment, 889-S1 cells ($5 \times 10^5$) were subcutaneously injected to the left and right flanks of male mice. Anti-PD-1 (BE0273, BioXcell) and anti-IgG isotype control (BE0089, BioXcell) were given intraperitoneally 100 μg per mouse twice a week from day 18 after tumour cell inoculation. The tumour progression was monitored two times per week using callipers.

**Single-cell preparation and flow cytometry.** Tumours or saline-perfused mouse lungs were mechanically minced with scissors to small pieces, followed by digestion at 37 °C in a 5% $CO_2$ incubator for 30 min in RPMI 1640 mixing with 10% FBS, penicillin/streptomycin, 0.05 mg/mL collagenase type I (C5894, Sigma-Aldrich), 0.05 mg/mL collagenase type IV (C1889, Sigma-Aldrich), 0.025 mg/mL hyaluronidase (H4272, Sigma-Aldrich), and 0.01 mg/mL DNase I (11284932001, Roche). Digested samples were passed through a 70 μM mesh, and red blood cells were lysed in RBC lysis buffer (420301, BioLegend) for 3 min. For cytokine staining, cells were firstly incubated with Ionomycin (500 ng/mL; S1672, Beyotime Biotechnology), PMA (5 ng/mL; P1585, Sigma-Aldrich), and Brefeldin A (1: 1000; 00-4506-51, eBioscience) at 37 °C for 4 h in a $CO_2$ incubator. Cells were then stained with fixable viability dye eFluor450 (65-0863-14, eBioscience) for 30 min, followed by Fc block with purified rat anti-mouse CD16/CD32 (553140, BD Biosciences). Cell labelling was performed with fluorescently conjugated antibodies directed against mouse CD45, CD11b, F4/80, CD11c, Gr-1, MHC II, Ly6G, PD-L1, CD31, CD3, CD8, CD4, PD-1, CD44, CD69, NK1.1, B220, Foxp3, CD25, IFN-γ, TNF-α or MHC I. Intracellular staining was performed using the Fixation/Permeabilization Solution Kit (eBioscience). Flow cytometry was carried out on LSRFortessa™ cell analyzer (BD Biosciences), and subsequent data were analysed using FlowJo X 10.0.7r2.

**Single-cell clones.** Single LLC cells were stringently gated, separated using single-cell separation modes, and robotically plated into a 96-well plate with 200 μL medium per well by a MoFlo XDP (Beckman Coulter, Fullerton, CA). After 1–2 weeks in culture, single-cell-derived colonies were transferred into 24-well plates for expansion.

**Luciferase assays.** Cells were cultured in 12 well plates, and then transiently transfected with NF-κB luciferase reporter plasmid (1 μg) and Renilla plasmid (100 ng). The cells were lysed at 48 h post infection and the reporter activities were monitored using the Dual-Luciferase Reporter Gene Assay Kit (RG027, Beyotime Biotechnology) on GloMax ® 20/20 Luminometer (Promega). Cells transfected with the USP12 promoter plasmids (1 μg) and Renilla plasmids (100 ng) were cultured for 24 h and then treated with API-2 or Rapamycin. After 24 h, cells were harvested for determining luciferase activity.

**ELISA and cytokine arrays.** Cells were cultured for 24 or 72 h. The cytokine concentrations in the cell supernatants were measured via ELISAs or human cytokine array, according to the manufacturer's instructions. Infinite® M1000 Pro (Tecan) was used to measure absorbance. The following ELISA kits were used in this study: mouse MCP-1/CCL2 ELISA kit (88-7391-22, Invitrogen), mouse CXCL1/KC ELISA kit (DY453, R&D Systems), human CXCL1/GRO alpha ELISA kit (DY275, R&D Systems), and human CXCL8 ELISA kit (88-8086-88,

Invitrogen). The human cytokine array (ARY005) was from R&D Systems (Minneapolis, MN).

**Monocyte differentiation and BMDM migration.** Supernatants were generated from LLC or 889-DTC cells cultured for 24 h. BMDMs were obtained from wild-type C57BL/6 mice as described[59]. In brief, bone marrow cells from the femurs were cultured in DMEM supplemented with 10% FBS and 20 ng/mL GM-CSF (315-03, PeproTech) and differentiated for 7 days. BMDMs were then cultured at $1 \times 10^6$ cells/mL in 6-well plates with LLC or 889-DTC supernatants; 48 h later the differentiation state was determined via qRT-PCR or flow cytometry. For migration assays, BMDMs ($1–2 \times 10^5$) were added in the top of 24-well chambers with inserts (8 μm pores; BD Biosciences), and conditioned media from LLC or 889-DTC cells were placed in the lower chamber. The plates were incubated for 22 h at 37 °C in a 5% $CO_2$ incubator. The migratory cells were fixed and stained with crystal violet.

**Immunohistochemistry (IHC) staining.** Slides were de-paraffinized, rehydrated, immersed in 3% hydrogen peroxide solution for 15 min, and boiled in a microwave for 20 min in 10 mM citrate buffer (pH 6.0). IHC was performed using a Ready-to-use IHC kit (Biotin free) (K405-50, Biovision) according to the manufacturer's instructions. The primary antibodies against USP12, PPM1B, and CD31 were used. The whole slide imaging was captured by Leica APERIO CS2, and image analysis was performed using Aperio ImageScope software v12.3.3.5048 (Leica Microsystems).

**Real-time reverse-transcription polymerase chain reaction (RT-PCR).** Total RNA was extracted from cells or tissues with RNAiso Plus (9108, Takara) and converted to cDNA using the PrimeScript™ RT Master Mix (RR036A, Takara) according to the manufacturer's recommendations. Quantitative real-time PCR was performed on a ViiA™ 7 Real-Time PCR System (Applied Biosystems) using Hieff® qPCR SYBR Green Master Mix (11202, YEASEN). All samples were normalized to GAPDH expression. The primers used are listed in Supplementary Table 3.

**RNA-sequencing (RNA-seq) analysis.** Total RNA from cells was isolated using TRIzol Reagent (15596018, Invitrogen), and purified using RNAClean XP Kit (A63987, Beckman Coulter) and RNase-Free DNase Set (79254, QIAGEN). RNA-seq studies were performed by Shanghai Biotechnology Corp (Shanghai, China). RNA libraries were made using VAHTS mRNA-seq V2 Library Prep Kit for Illumina (NR601-02, Vazyme) (for MEFs) or VAHTS Stranded mRNA-seq Library Prep Kit for Illumina (NR602-02, Vazyme) (for A549 and H1944 cells), and then submitted to Hiseq 2500 system (Illumina). Adaptor trimming and rRNAs filtering were achieved by Seqtk (https://github.com/lh3/seqtk). The clean reads were mapped to the mouse reference genome (GRCm38.p4) for MEFs, or the human reference genome (GRCh38) for human cells, using Hisat2 (version 2.0.4) to generate the BAM files for each sample. The uniquely mapped fragments of each gene were counted by Stringtie (version 1.3.0). The raw data of RNA-seq and detailed experimental design are deposited under GEO numbers GSE156958, GSE156959, and GSE174078. Differentially expressed genes in RNA-seq data were determined by edgeR software (version 3.30.3). The pathway enrichment analysis and GSEA analysis were performed using the ClusterProfiler package (version 3.16.1)[60] through R programming language (version 3.6.2).

**Western blotting and immunoprecipitation assays.** For IP-ubiquitination assay, cells were lysed with RIPA buffer (P0013K, Beyotime Biotechnology) containing 1% Triton X-100, 1% sodium deoxycholate, 0.1% SDS, the protease inhibitor (05892970001, Roche Diagnostics), and phosphatase inhibitor cocktail (04906845001, Roche Diagnostics). For co-immunoprecipitation (co-IP), cells were lysed with Western and IP lysis buffer (P0013, Beyotime Biotechnology) containing the protease and phosphatase inhibitors. The supernatants of cell lysates were incubated with Protein G-agarose suspension (16-266, Millipore) together with specific antibodies. After being incubated overnight, the precipitates were washed three times with cold IP buffer and boiled with loading buffer for western blotting. Images were visualized by the ChemiDoc™ XRS system (Bio-Rad) or Odyssey Sa Infrared Imaging System (LI-COR). The band intensity of western blotting was analysed by Odyssey Sa Imaging System Application Software (Version 1.1.7).

**Mass spectrometry analysis.** Purified proteins using Flag M2 affinity gel (A2220, Sigma-Aldrich) were separated by SDS-PAGE followed by stained coomassie blue. The entire lane was excised, digested with trypsin (Promega), and then subjected to LC-MS/MS analysis performed by Applied Protein Technology (APTBIO, Shanghai, China). Briefly, peptide samples were injected into the Easy-nLC 1000 system (Thermo Fisher, California, USA), loaded by Acclaim™ PepMap™ 100 (100 μm × 2 cm, nanoViper C18; Thermo Scientific), separated by Thermo scientific EASY column (10 cm, ID75 μm, 3 μm, C18-A2) at 300 nL/min for 60 min, and eluted with three-step acetonitrile (0.1% formic acid in 84% acetonitrile) gradient: 0–35% over the first 50 min, 35–100% for 50–55 min, and 100% for 55–60 min. The tandem mass spectrometry was performed by Q Exactive mass spectrometer (Thermo Fisher, California, USA). MS RAW data were processed

using Biopharma Finder1.0 software for protein identification with the following settings: mass values, monoisotopic; fixed modifications, carbamidomethyl (C); variable modifications, oxidation (M); protein and peptide FDR, ≦ 0.01; peptide mass tolerance, ±20 ppm; max missed cleavages, 2.

**Bioinformatics analyses of clinical data**. The bioinformatics analyses were completed through R programming language (version 3.6.2) or GraphPad Prism 5 software (GraphPad Software). The FPKM-normalized RNA-seq data of TCGA datasets were downloaded using TCGAbiolinks package (version 2.16.4)[61] and then were transformed to TPM values for further calculation. The transcriptome databases from the GEO database, including GSE31210, GSE30219, GSE19804, and GSE10072, were downloaded by GEOmirror package (version 0.1.0), in which the NSCLC samples were selected for further analysis. Spearman correlation analyses were performed by GraphPad Prism 5 software (GraphPad Software). Differentially expressed genes between tumour and non-tumour tissue samples were calculated using DESeq2 package (version 1.28.1) and visualized by Volcano plots generated by ggplot2 package (version 3.3.3). Cutoff values for the tested factors in survival analyses were estimated by maxstat package (version 0.7–25), and the Kaplan-Meier curves were generated by GraphPad Prism 5 software. GSEA analysis was performed using Clusterprofiler package (version 3.16.1). The abundance of tumour-infiltrating macrophage in tumour tissues was estimated by xCell (version 1.1.0) method[24]. The predicted clinical response to immune checkpoint blockade (ICB) therapy of each patient in TCGA-LUAD and GSE30219 databases was predicted by the Tumour Immune Dysfunction and Exclusion (TIDE) algorithm (tidepy-1.3.3.1) with the defaulted parameters[28].

**Statistics and reproducibility**. Before performing statistical analysis, Shapiro-Wilk test was used to detect the data distribution. Then, a two-sided Student's t-test or Mann-Whitney U-test was used for comparison of two groups, and a parametrical or non-parametrical ANOVA test was used for multiple-comparison experiments. All of these analyses were done using IBM SPSS Statistics 23 software (IBM Corporation) or GraphPad Prism 5 software (GraphPad Software). The significance of enrichment analysis was calculated by over-representation test algorithm and the P-value was adjusted by Bonferroni method, which was implemented in clusterProfiler package. The permutation test was used to calculate the nominal P for GSEA. Other statistical methods are indicated in the legends. Results are performed at least three independent experiments for in vitro analyses. For western blotting analysis, the images are representative of at least two independent experiments. For in vivo experiments, at least two independent replicates are generated. One representative experiment is shown. A P value < 0.05 was considered to be statistically significant.

**Ethics statement**. All NSCLC samples were collected in Ren Ji Hospital, Shanghai Jiao Tong University School of Medicine (Shanghai, China) (Supplementary Table 4). All patients provided written informed consent for sampling and research. This study was approved by the Ren Ji Hospital Ethics Committee. All animal procedures were approved by the Institutional Animal Care and Use Committee of Shanghai Jiao Tong University and Shanghai Cancer Institute.

**Reporting summary**. Further information on research design is available in the Nature Research Reporting Summary linked to this article.

## Data availability

Mass Spectrum data that supporting our findings have been deposited in MassIVE under the accession code MSV000087785. The RNA-seq data generated in this study have been deposited in the NCBI gene expression omibus (GEO) under accession code GSE156958, GSE156959, and GSE174078. TCGA datasets used in this study are available in NCI's Genomic Data Commons (GDC) and were accessed using TCGA biolinks package. The publicly GEO datasets used in this study are available in the GEO database under accession code GSE31210, GSE30219, GSE19804, and GSE10072. The remaining data are available within the Article, Supplementary Information or Source Data file. Source data are provided with this paper.

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

## Acknowledgements

This work was supported by the National Natural Science Foundation of China (81972579, 81572694, 81802746), Shanghai Natural Science Fund (20ZR1454100).

## Author contributions

Z.Y. and Y.Z.L. defined the project and designed the experiments. Z.Y., G.X. and B.W. carried out most of the experiment and analysed data. Y.L., L.Z., T.J., M.T., X.X., K.J. and L.X. participated in experiment conduction. Z.Y. and B.W. performed bioinformatics analysis. Y.F., D.T., X.Z., G.Z. and W.J. provided technical or material support. Z.Y. and Y.Z.L. wrote and edited the paper. X.Z. and Y.Z.L. jointly supervised the study.

## Competing interests

The authors declare no competing interests.
