## [Peer Review File · Nature Communications]

REVIEWER COMMENTS

Reviewer #1 (Remarks to the Author):

In this study authors investigate the role of USP12 in lung cancer and report that AKT-mTOR downregulates USP12 which generates oncogenic conditions. They identify PPM1B as a novel target for USP12 implicating USP12 in the NF-KB signalling pathway.

This manuscript is very well written and presented. It is a pleasure to read. Findings are generally clear and supported by the data. This work should be of interest to the scientific community at large. There are few specific comments that warrant some additional attention though.

1. Authors propose that USP12 deubiquitinates PPM1B. However it is not clear if the immunoprecipitation presented in the manuscript to demonstrate this was done under denaturing conditions. This is absolutely critical as otherwise it is very likely that authors are seeing the changes in the ubiquitination status of PPM1B interacting proteins rather than just PPM1B itself. This needs to be clarified in the methods section and repeated under denaturing conditions if it hasn't been. This is the most critical point of my review.

2. Authors focused on few cytokines for the NF-KB signalling analysis. It would be helpful to broaden this panel and preferably to perform an unbiased RNAseq of USP12 silenced cells compared to WT to assess if NF-KB signalling is truly affected and if it is the key affected pathway. However, if money prevents this qPCR could be employed.

3. Authors propose that mTOR-AKT pathway downregulates USP12 levels. However, no mechanism for this observation is proposed. As this is quite an important part of the manuscript mechanistic explanation feels necessary especially as USP12 itself was previously found to regulate AKT phosphorylation via targeting PHLPP and PHLPL. This would imply a feedback loop hence mechanism needs to be proposed.

4. All small sample in vitro data seems tested with parametric statistics. However, there is no indication in the methods or in the legends if the data has ever been tested for normal distribution. It is likely that this data is not parametric (due to small sample sizes) and this needs to be tested before parametric statistics are applied. Text should also reflect this.

5. To validate if the effects are truly USP12 driven it would be worth looking at USP46 (>80% homology and almost complete overlap of target proteins reported so far) and also to silence WDR48, (USP12, USP46 and USP1 required partner) as this would validate the deubiquitination side of the story.

6. Why has ANOVA been used in Fig 2D? It does not seem to be the appropriate test.

7. How common is significant and meaningful USP12 downregulation in clinical lung patient samples? This needs to be considered in light of USP12 being found as one of the 12 most commonly overexpressed cancer-associated genes located near an amplified super-enhancer (Zhang X, Choi PS, Francis JM, Imielinski M, Watanabe H, Cherniack AD, et al. Identification of focally amplified lineage-specific super-enhancers in human epithelial cancers. *Nat Genet.* 2016; 48: 176–82). Is lung an outlier here?

Reviewer #2 (Remarks to the Author):

In the manuscript from Yang et al., the authors mine experimental profiling data and public patient datasets for deubiquitinases that are dysregulated and may be involved in regulating lung tumor progression. They identify USP12 and demonstrate that in syngeneic or autochthonous KRAS mutant murine models its expression regulates tumor progression. They further explore the mechanistic basis

for this and identify that regulation of the chemokine expression results in an altered immune microenvironment. Overall, the studies are robust and performed extremely well, with appropriate consideration of controls. I find the work innovative and conceptually provocative. The manuscript provides an important advance in our basic understanding of factors that regulate the tumor microenvironment and that can affect the response to immune checkpoint inhibitor therapy. I have several concerns that should be addressed before the manuscript can be considered for publication. Specific comments and suggestions:

1. Data in Fig. 1 does not break down the analyses of human lung tumors (e.g., Fig. 1e-g) into any of the relevant subgroups, including no consideration of histology, driver oncogenotype, or stage. The presented data certainly support the findings that USP12 is downregulated in NSCLC, but I am not clear if this is a phenomena linked to specific oncogenotypes of NSCLC (like KRAS mutant), or if it occurs more broadly. In this sense the missing data are critical to truly understanding how a therapeutic approach might be translated into the clinic—broadly for all patients or more selectively for those with particular driver mutations.
2. Similarly, there needs to be histopathologic evaluation of the spontaneous lung lesions in the animals in Fig. 1C for type and grade. Since mutant KRAS only tumors are primarily adenomas, it is not clear to me if the same effect occurs from loss of USP12 in very early lesions and in more invasive adenocarcinomas. Again, these details are important to understand both the biology of USP12 and whether the animal models are faithful models of a specific human disease type or subtype.
3. The authors do not explore the tumor cell autonomous effects of USP12 and whether the observed phenotypes are entirely due to regulating the tumor immune microenvironment. Implantation of the LLC and 889-S1 tumors into immunocompromised mice would help to understand if this DUB also has tumor cell intrinsic effects along with the regulation of chemokine/cytokine secretion.
4. The data presented in Fig. 5 does not clearly define a causal link between the TAM population and suppression of the cytotoxic T cell populations in the tumors. The data could be interpreted to suggest that altered chemokine secretion by tumor cells directly modulates cytotoxic T cells and TAMs simultaneously. But it seems that the authors are suggesting a direct effect of tumors on TAMs, and that the TAMs suppress the T cells. The authors should clarify their explanation and model (e.g. the schema in Fig. 6C does not delineate one model vs the other). If they want to propose an effect of TAMs on the T cell biology, then this will require additional experimental evidence to justify the conclusion, such as a depletion study demonstrating the effect of TAMs.
5. It is not clear how the authors have performed the analyses presented in Fig. 6B. Since the public datasets cited do not have treatment outcomes, it needs to be better explained how they have coded some samples as corresponding to responders and others to non-responders.

Minor comment:

1. Line 143, "lentinivirally" is misspelled.

Reviewer #3 (Remarks to the Author):

The authors identified USP12 downregulation in human and mouse lung cancer and further showed that USP12 downregulation creates an immune-suppressive tumor microenvironment in part through PD-L1 upregulation. At the molecular level, USP12 protects from ubiquitination-mediated degradation of PPM1B, a suppressor of NFκB. Overall this is an outstanding study with the main conclusion largely supported by their data. Few concerns include:

1. It has been shown that USP12 expression is upregulated in multiple types of tumors including lung cancer (Nat Genet, 2016 Feb; 48(2): 176-82), this discrepancy needs to be discussed.
2. The functional consequences of USP12-mediated PPM1B deubiquitination should be further validated by pulse chase analysis.
3. It will be interesting to further look at whether PD-L1 expression is reversely correlated with USP12 in human lung cancers.
4. A variety of transcription factors are involved in PD-L1 expression, data to show whether USP12 inhibits PD-L1 expression through NF-κB should be provided. A simple experiment is to test whether a NF-κB inhibitor diminishes PD-L1 upregulation in USP12 knockdown cells.
5. Treg analysis is not convincing, analysis with additional markers such as CD25 in addition to FoxP3 on the gated CD4 T cells should be used.

Reviewer #1 (Remarks to the Author):

In this study authors investigate the role of USP12 in lung cancer and report that AKT-mTOR downregulates USP12 which generates oncogenic conditions. They identify PPM1B as a novel target for USP12 implicating USP12 in the NF-KB signalling pathway.

This manuscript is very well written and presented. It is a pleasure to read. Findings are generally clear and supported by the data. This work should be of interest to the scientific community at large.

There are few specific comments that warrant some additional attention though.

We appreciate very much the positive comments and great efforts of the reviewer to improve the quality of our paper. Below is a point-by-point reply to the reviewer.

1. Authors propose that USP12 deubiquitinates PPM1B. However it is not clear if the immunoprecipitation presented in the manuscript to demonstrate this was done under denaturing conditions. This is absolutely critical as otherwise it is very likely that authors are seeing the changes in the ubiquitination status of PPM1B interacting proteins rather than just PPM1B itself. This needs to be clarified in the methods section and repeated under denaturing conditions if it hasn't been. This is the most critical point of my review.

The concern of the reviewer is important. We should have described the method more clearly in the previous version of our manuscript. The immunoprecipitation for measuring protein ubiquitination needs to be done in a denature condition to avoid possible contamination derived from other ubiquitinated proteins that may interact with the target to be checked. In our system, the cells used for IP-ubiquitination assay were lysed with the denaturing buffer containing 1% Triton X-100, 1% sodium deoxycholate and 0.1% SDS, and protein interaction in the lysates can be blocked. We have added the details of the immunoprecipitation assay in the Methods section in the

new version of our manuscript.

2. Authors focused on few cytokines for the NF-KB signalling analysis. It would be helpful to broaden this panel and preferably to perform an unbiased RNAseq of USP12 silenced cells compared to WT to assess if NF-KB signalling is truly affected and if it is the key affected pathway. However, if money prevents this qPCR could be employed.

Following the suggestion of the reviewer, we performed RNAseq experiments to profile the transcriptional patterns of lung tumour cells with or without the silencing of USP12. GSEA analysis revealed an enhanced activation in the transcription of the NF- κ B target genes in H1944 cells with USP12-knockdown, indicating that USP12 negatively regulates NF- κ B signalling (Supplementary Fig. 3a). The information has been added in the sections of Methods and Data availability.

3. Authors propose that mTOR-AKT pathway downregulates USP12 levels. However, no mechanism for this observation is proposed. As this is quite an important part of the manuscript mechanistic explanation feels necessary especially as USP12 itself was previously found to regulate AKT phosphorylation via targeting PHLPP and PHLPL. This would imply a feedback loop hence mechanism needs to be proposed.

We appreciate the reviewer for the suggestion. Our previous results indicate that the AKT-mTOR pathway suppresses USP12 transcription. In the revision, we further validated the results with USP12-promoter luciferase experiments, and found that both the AKT inhibitor (API-2) and the mTOR inhibitor (Rapamycin) significantly increased the luciferase activities driven by the USP12 promoter (Supplementary Fig. 1f, g). Serial deletion analysis of the transcriptional activities of the USP12 promoter showed that the region -2928/-1337 within the USP12 promoter was important for the AKT-mTOR-mediated upregulation (Supplementary Fig. 1h). The cis-elements within

the region responsible for USP12 downregulation by AKT-mTOR signalling and the related biochemical details warrant further study. To test whether there is a feedback regulation between USP12 and AKT, we examined the effects of USP12 overexpression on AKT activation. Unexpectedly, our results showed that USP12 expression in lung tumour A549 and H1944 cells did not decrease phosphorylated AKT levels (Supplementary Fig. 1i, j), suggesting a context-dependant effects of USP12 on AKT activation in different scenarios.

4. All small sample in vitro data seems tested with parametric statistics. However, there is no indication in the methods or in the legends if the data has ever been tested for normal distribution. It is likely that this data is not parametric (due to small sample sizes) and this needs to be tested before parametric statistics are applied. Text should also reflect this.

Following the reviewer's suggestion, we have performed Shapiro-Wilk test to check the distribution of our data. We then used two-sided Student's *t*-test or Mann-Whitney U-test for comparison of two groups, and parametrical or nonparametrical ANOVA tests for multiple-comparison experiments. According the results of Shapiro-Wilk test, the figures (Fig. 2d; Fig. 3d, e; Fig. 5f; Fig. 6a; Supplementary Fig. 2c-e; and Supplementary Fig. 3b, c) have been modified. The statistical methods and the figure legends have been modified in our revised manuscript.

5. To validate if the effects are truly USP12 driven it would be worth looking at USP46 (>80% homology and almost complete overlap of target proteins reported so far) and also to silence WDR48, (USP12, USP46 and USP1 required partner) as this would validate the deubiquitination side of the story.

Thanks for the reviewer's thoughtful suggestions. We evaluated whether USP46 could interact with PPM1B by IP experiments. The results showed that USP46 was capable

of interacting with PPM1B (Supplementary Fig. 3f), and that USP46 overexpression marginally increased levels of PPM1B protein (Supplementary Fig. 3g), whereas silencing USP46 resulted in a slight decrease in PPM1B expression (Supplementary Fig. 3h).

Although these results indicate that USP46 may regulate PPM1B expression to some extent in cultured cells, clinical data from the transcriptional datasets of NSCLC revealed that USP12 but not of USP46 was downregulated in tumour specimens compared with normal tissues (Supplementary Fig. 3i and Fig. 1e, f in the manuscript), indicating that downregulation of USP12 accounts for the impairment in PPM1B expression in human NSCLC samples. In addition, the abundancy of USP46 transcripts in tumours was apparently lower in comparison with USP12 mRNA levels (Supplementary Fig. 3j). We also examined PPM1B expression in cells with WDR48 knock-down, and, in line with the previous recognition of WDR48 as the partners of USP12, we found that silencing either WDR48 or USP12 yielded a similar inhibition of PPM1B expression (Supplementary Fig. 3e). Collectively, in the perspective of clinical relevance of USP12 downregulation in NSCLC, these data support our conclusion that dysregulation of the USP12-mediated deubiquitination is an important event in the development of an immune-suppressive environment in NSCLC.

6. Why has ANOVA been used in Fig 2D? It does not seem to be the appropriate test.

We thank the reviewer very much for the correction. We recalculated statistical differences between the tumour sizes of two groups with Mann-Whitney U-test, and the data have been revised in the manuscript.

7. How common is significant and meaningful USP12 downregulation in clinical lung patient samples? This needs to be considered in light of USP12 being found as one of the 12 most commonly overexpressed cancer-associated genes located near an amplified super-enhancer (Zhang X, Choi PS, Francis JM, Imielinski M, Watanabe H, Cherniack AD, et al. Identification of focally amplified lineage-specific

super-enhancers in human epithelial cancers. Nat Genet. 2016;48:176–82). Is lung an outlier here?

We carefully read the paper published in Nat Genetics (2016; 48:176-82). Basically, the authors described a genome-wide picture of amplification of the super-enhancers in multiple cancer types. By screening 12 types of tumours with somatic copy number analysis and H3K27ac ChIP-seq data, they identified six focally amplified super-enhancers, which resulted in upregulated transcription of six genes, in different types of tumours. Among them, the amplification of 21-kb region (chr. 13: 27,523,026–27,544,353), which contains the super-enhancer for USP12 transcriptional activation, was found in colorectal carcinoma (CRC) and named **USP12-CCSE (USP12 colorectal carcinoma super-enhancer**, Fig. 1B and D in that paper). Of note, USP12-CCSE amplification was not mentioned or found in other types of tumours studied in that paper. More importantly, even in CRCs, the frequency of the amplification of USP12-CCSE is very low based on the data of Fig. 1D (among the tumours checked, 6 CRC tumours with focal amplification of USP12-CCSE alone and 127 tumours without the amplification) and the data in the Suppl table. 1. Therefore, while the regulation mediated by USP12-CCSE has been appreciated as a mechanism for USP12 activation in CRC, the event is not commonly present in CRC, and, presumably, USP12-CCSE amplification may be a rare occurrence, if not absent, in other types of tumours. We think that the biological significance and clinical relevance of USP12 in colorectal cancer are still unclear and may need to be addressed with experimental data.

Reviewer #2 (Remarks to the Author):

In the manuscript from Yang et al., the authors mine experimental profiling data and public patient datasets for deubiquitinases that are dysregulated and may be involved in regulating lung tumor progression. They identify USP12 and

demonstrate that in syngeneic or autochthonous KRAS mutant murine models its expression regulates tumor progression. They further explore the mechanistic basis for this and identify that regulation of the chemokine expression results in an altered immune microenvironment. Overall, the studies are robust and performed extremely well, with appropriate consideration of controls. I find the work innovative and conceptually provocative. The manuscript provides an important advance in our basic understanding of factors that regulate the tumor microenvironment and that can affect the response to immune checkpoint inhibitor therapy. I have several concerns that should be addressed before the manuscript can be considered for publication.

We thank the reviewer for the positive comments, and the concerns and suggestions are truly helpful to improve the quality of our paper. Below is a point-by-point reply to the reviewer.

Specific comments and suggestions:

1. Data in Fig. 1 does not break down the analyses of human lung tumors (e.g., Fig. 1e-g) into any of the relevant subgroups, including no consideration of histology, driver oncogenotype, or stage. The presented data certainly support the findings that USP12 is downregulated in NSCLC, but I am not clear if this is a phenomena linked to specific oncogenotypes of NSCLC (like KRAS mutant), or if it occurs more broadly. In this sense the missing data are critical to truly understanding how a therapeutic approach might be translated into the clinic—broadly for all patients or more selectively for those with particular driver mutations.

This is a great suggestion. To study whether USP12 downregulation is strictly associated with *KRAS* mutation in NSCLC, we analysed USP12 transcript levels in human LUADs carrying different types of oncogenic mutations (TCGA-LUAD database). We found that decreased expression of USP12 was also present in tumours with other driver gene mutations (Fig. 1e), indicating that the downregulation of

USP12 was not an event specifically present in *KRAS*-mutant NSCLC. The phenomenon is actually supported by the data that the AKT-mTOR signalling, which acts downstream of the mutant drivers, is responsible for USP12 downregulation. In addition to human LUAD, low expression of USP12 was found in tumours of LUSCs (Supplementary Fig. 1c). Moreover, we found that the USP12 expression was downregulated in the tumours of the patients at different stages (Supplementary Fig. 1d). Together, these results indicate that USP12 downregulation is a common phenomenon in NSCLC.

2. Similarly, there needs to be histopathologic evaluation of the spontaneous lung lesions in the animals in Fig. 1C for type and grade. Since mutant KRAS only tumors are primarily adenomas, it is not clear to me if the same effect occurs from loss of USP12 in very early lesions and in more invasive adenocarcinomas. Again, these details are important to understand both the biology of USP12 and whether the animal models are faithful models of a specific human disease type or subtype.

The suggestion of the reviewer is important. According the research of Tyler Jacks et al., Grade 3 adenocarcinomas can develop in *Kras*^{G12D}-driven lung tumour mouse model (Cancer Res. 2005 Nov 15;65(22):10280-8; Nat Protoc. 2009;4(7):1064-72), though the frequency of the lesions is low. We examined USP12 expression in the *Kras*^{G12D} mouse lung tumour containing Grade 2 and 3 lesions, and found that the downregulation of USP12 was also present in tumour cells of the adenocarcinoma lesions (shown below). The results are generally consistent with the data of human NSCLCs (Supplementary Fig. 1d).

3. The authors do not explore the tumor cell autonomous effects of USP12 and whether the observed phenotypes are entirely due to regulating the tumor immune microenvironment. Implantation of the LLC and 889-S1 tumors into immunocompromised mice would help to understand if this DUB also has tumor cell intrinsic effects along with the regulation of chemokine/cytokine secretion.

The comment is very important. We performed more experiments in immune-competent and immunocompromised mice. We found that overexpression of USP12 but not of USP12-C48S significantly inhibited LLC and 889-S1 tumour growth in immune-competent mice, whereas the inhibitory effects of USP12 did not display in immune-deficient mice (B-NDG mice, *NOD-Prkdc^{scid}IL2rg^{tm1}/Bcgen*) (Supplementary Fig. 2g, h). These results strongly indicate that USP12 regulates tumour growth in a non-tumour cell autonomous manner.

4. The data presented in Fig. 5 does not clearly define a causal link between the TAM population and suppression of the cytotoxic T cell populations in the tumors. The data could be interpreted to suggest that altered chemokine secretion by tumor cells directly modulates cytotoxic T cells and TAMs simultaneously. But it seems that the authors are suggesting a direct effect of tumors on TAMs, and that the

TAMs suppress the T cells. The authors should clarify their explanation and model (e.g. the schema in Fig. 6C does not delineate one model vs the other). If they want to propose an effect of TAMs on the T cell biology, then this will require additional experimental evidence to justify the conclusion, such as a depletion study demonstrating the effect of TAMs.

The concern of the reviewer is thoughtful. We cannot exclude the possibility that the altered production of the chemokines caused by USP12 downregulation in tumour cells may have effects on T cells and TAMs simultaneously. For instance, previous studies have revealed the suppressive function of CCL2 and CCL5 on T cell activity and the inhibitory effects of CXCL1 on T cell infiltration into tumours (Immunol Lett. 2003 Dec 15;90(2-3):187-94; Immunol Lett. 2004 Feb 15;91(2-3):239-45; Immunity. 2018 Jul 17;49(1):178-193). There are a number of chemokines affected by USP12, and the combinatory effects of the aberrantly expressed chemokines on growth and ICB response of tumours with USP12 downregulation are possibly fulfilled through regulating different types of immune cells. This is reasonable and we have discussed the point in our revised manuscript.

5. It is not clear how the authors have performed the analyses presented in Fig. 6B. Since the public datasets cited do not have treatment outcomes, it needs to better explained how they have coded some samples as corresponding to responders and others to non-responders.

This is a great suggestion. The NSCLC datasets used do not include the information of treatment outcomes. TIDE (<http://tide.dfci.harvard.edu/>), a computational method, has been used to predict the ICB response (Nat Med. 2018 Oct;24(10):1550-1558; Cancer Discov. 2021 Feb 15;candisc.0812). By extracting transcription profiles of pretreatment tumours and integration analysing expression signatures of T-cell dysfunction and T-cell exclusion, TIDE can predict the extent to which tumour immune tolerance has been established. Therefore, we utilized TIDE to predict ICB

response and non-response of the patients based on the NSCLC datasets.

Minor comment:

1. Line 143, “lentinvirally” is misspelled.

We apologize for the mistake, and we corrected the word in our revised manuscript.

Reviewer #3 (Remarks to the Author):

The authors identified USP12 downregulation in human and mouse lung cancer and further showed that USP12 downregulation creates an immune-suppressive tumor microenvironment in part through PD-L1 upregulation. At the molecular level, USP12 protects from ubiquitination-mediated degradation of PPM1B, a suppresser of NFkB. Overall this is an outstanding study with the main conclusion largely supported by their data. Few concerns include:

We appreciate very much the positive comments and great questions of the reviewer that help revise our paper. Below is a point-by-point reply to the reviewer.

1. It has been shown that USP12 expression is upregulated in multiple types of tumors including lung cancer (Nat Genet, 2016 Feb;48(2):176-82), this discrepancy needs to be discussed.

The concern is important, and Reviewer#1 also raised the question. We read through the paper published in Nat Genetics (2016; 48:176-82), which described a genome-wide picture of amplification of the super-enhancers in multiple cancer types. By screening 12 types of tumours with somatic copy number analysis and H3K27ac ChIP-seq data, they identified six focally amplified super-enhancers, which resulted in upregulated transcription of six genes, in different types of tumours. Among them, the

amplification of 21-kb region (chr. 13: 27,523,026–27,544,353), which contains the super-enhancer for USP12 transcriptional activation, was found in colorectal carcinoma (CRC) and named **USP12-CCSE (USP12 colorectal carcinoma super-enhancer)**, Fig. 1B and D in that paper). Of note, USP12-CCSE amplification was not mentioned or found in other types of tumours studied in that paper. More importantly, even in CRCs, the frequency of the amplification of USP12-CCSE is very low based on the data of Fig. 1D (among the tumours checked, 6 CRC tumours with focal amplification of USP12-CCSE alone and 127 tumours without the amplification) and the data in the Suppl table. 1. Therefore, while the regulation mediated by USP12-CCSE has been appreciated as a mechanism for USP12 activation in CRC, the event is not commonly present in CRC, and, presumably, USP12-CCSE amplification may be a rare occurrence, if not absent, in other types of tumours. In our opinion, the biological significance and clinical relevance of USP12 in colorectal cancer are still unclear and may need to be addressed with more experimental data.

2. The functional consequences of USP12-mediated PPM1B deubiquitination should be further validated by pulse chase analysis.

Thanks for the thoughtful suggestion of the reviewer. We examined the effect of USP12 on the stability of PPM1B in the presence of cycloheximide (CHX) (Supplementary Fig. 3d). The results indicate that USP12 is capable of increasing PPM1B stability.

3. It will be interesting to further look at whether PD-L1 expression is reversely correlated with USP12 in human lung cancers.

Following the suggestion of the reviewer, we analysed the correlation between USP12 and CD274 in the human NSCLC datasets. The results revealed a negative correlation between USP12 and PD-L1 transcript levels in NSCLCs (Supplementary Fig. 7g).

4. A variety of transcription factors are involved in PD-L1 expression, data to show whether USP12 inhibits PD-L1 expression through NF- κ B should be provided. A simple experiment is to test whether a NF- κ B inhibitor diminishes PD-L1 upregulation in USP12 knockdown cells.

We used the NF- κ B inhibitors, pyrrolidinedithiocarbamate ammonium (PDTC, 20 μ M), Bay 11-7082 (5 μ M), and BMS-345541(10 μ M), to identify whether USP12 regulation of PD-L1 expression involves NF- κ B pathway. However, NF- κ B inhibition in cultured tumour cells did not diminish PD-L1 upregulation caused by USP12-knockdown. The results are shown below. Of note, previous studies also reveal that NF- κ B signalling in some conditions has no effect on PD-L1 expression (Cancer Discov. 2021 Feb 15; candisc.0812.2020; Sci Transl Med. 2020 Oct 14;12(565):eabb0152). Therefore, we speculate that other mechanism may underlie the modulation of PD-L1 expression by USP12.

5. Treg analysis is not convincing, analysis with additional markers such as CD25 in addition to FoxP3 on the gated CD4 T cells should be used.

Follow the reviewer's suggestion, we analysed the proportion of tumour infiltrating CD4⁺Foxp3⁺CD25⁺Tregs. The results showed that a higher fraction of CD4⁺Foxp3⁺CD25⁺ cells in shUSP12 tumours compared with tumours with shControl (Supplemental Fig. 5a). Gating strategy is also updated in Supplemental Fig. 4.

REVIEWERS' COMMENTS

Reviewer #1 (Remarks to the Author):

I would like to congratulate the authors on their thoughtful review which has further improved this great piece of work. My only suggestion would be to make the USP12 RNAseq available as a resource to the scientific community by either adding the full dataset as an Excel table in supplementary files or by uploading it to a public database. Also the methods section for the denaturing IP needs a quick proof-read.

Reviewer #2 (Remarks to the Author):

The authors have carefully and appropriately revised the manuscript to address my original critiques.

Reviewer #3 (Remarks to the Author):

All my concerns have been adequately addressed, the revised manuscript has been improved and carries important new insights regarding to tumor USP12 in antitumor immune response. Acceptance for publication is recommended.

Reviewer #1 (Remarks to the Author):

I would like to congratulate the authors on their thoughtful review which has further improved this great piece of work. My only suggestion would be to make the USP12 RNAseq available as a resource to the scientific community by either adding the full dataset as an Excel table in supplementary files or by uploading it to a public database.

Also the methods section for the denaturing IP needs a quick proof-read.

We are grateful to the reviewer for the suggestions. We have uploaded the datasets in the NCBI gene expression omnibus (GEO) under accession code GSE156958, GSE156959 and GSE174078 (the data generated in the 1st revision). We also revised the part of denaturing IP method.

Reviewer #2 (Remarks to the Author):

The authors have carefully and appropriately revised the manuscript to address my original critiques.

We appreciate very much the help of the reviewer for paper revision.

Reviewer #3 (Remarks to the Author):

All my concerns have been adequately addressed, the revised manuscript has been improved and carries important new insights regarding to tumor USP12 in antitumor immune response. Acceptance for publication is recommended.

We appreciate very much for the time the referee spent for reviewing our paper.